# Influence of covariance of aerosol and meteorology on co-located precipitating and non-precipitating clouds over Indo-Gangetic Plains

Nabia Gulistan[1], Khan Alam[1*], Yangang Liu[2]

[1]Department of Physics, University of Peshawar, Peshawar, 25120, Pakistan
[2]Environmental & Climate Science Department, Brookhaven National Laboratory, USA

**Correspondence**: Khan Alam (khanalam@uop.edu.pk)

## HIGHLIGHTS

- Strong aerosol-cloud relations under unstable meteorological conditions led to the formation of thick precipitating clouds.
- In thick clouds, the activation of cloud droplets is weakly dependent on aerosols.
- Optically thin clouds led to a high precipitation rate.

## ABSTRACT

Aerosol-cloud-precipitation-interaction (ACPI) plays a pivotal role in the global and regional water cycle and the earth's energy budget; however, it remains highly uncertain due to the underlying different physical mechanisms. Therefore, this study aims to systematically analyze the effects of aerosols and meteorological factors on ACPI in the co-located precipitating (PCs) and non-precipitating clouds (NPCs) clouds in winter and summer seasons by employing the long-term (2001-2021) retrievals from Moderate Resolution Imaging Spectroradiometer (MODIS), Tropical Rainfall Measuring Mission (TRMM) coupled with the National Center for Environmental Prediction/National Center for Atmospheric Research (NCEP/NCAR) reanalysis-II datasets over the Indo-Gangetic Plains (IGP). The results exhibit a decadal increase in aerosol optical depth (AOD) over Lahore (5.2%), Delhi (9%), Kanpur (10.7%), and Gandhi College (22.7%) and a decrease over Karachi (-1.9%) and Jaipur (-0.5%). The most stable meteorology with high values of lower tropospheric stability (LTS) is found in both seasons over Karachi. In the summer season, the occurrence frequency of clouds is high (74%) over Gandhi College, 60% of which are PCs. Conversely, the least number of PCs are found over Karachi. Similarly, in the winter season, the frequency of cloud occurrence is low over Karachi and high over Lahore and Gandhi College. The analysis of cloud top pressure (CTP) and cloud optical thickness (COT) indicate high values of cloud fraction (CF) for thick and high-level clouds over all study areas except Karachi. The microphysical properties such as cloud effective radius (CER) and cloud droplet number

concentration (CDNC) bear high values (CER $> \sim 15$ μm and CDNC $> \sim 50$ cm$^{-3}$) for both NPCs
and PCs in summer. The AOD-CER correlation is good (weak) for PCs (NPCs) in winter. Similarly,
the sensitivity value of the first indirect effect ( FIE ) is high (ranging from $0.2 \pm 0.13$ to $0.3 \pm 0.01$
in winter, and from $0.19 \pm 0.03$ to $0.32 \pm 0.05$ in summer) for PCs and low for NPCs. The
sensitivity value for the second indirect effect (SIE) is relatively higher (such as $0.6 \pm 0.14$ in
winter and $0.4 \pm 0.04$ in summer) than FIE. Sensitivity values of the aerosol-cloud interaction
(ACI) are low (i.e., $-0.06 \pm 0.09$) for PCs in summer Furthermore, the precipitation rate (PR)
exhibits high values in summer season, primarily due to the significant contribution from optically
thick clouds with lower CDNC ($< \sim 50$ cm$^{-3}$) and larger CER, and intermediate contribution from
optically thick clouds with higher CDNC ($> \sim 50$ cm$^{-3}$ ).

**Keywords:** Aerosol-cloud-precipitation-interaction, Aerosol optical depth, cloud effective radius,
cloud droplet number concentration, lower tropospheric stability, relative humidity, first indirect
effect, second indirect effect, precipitation sensitivity.

## 1. Introduction

The aerosol-cloud-precipitation-interaction (ACPI) and aerosol-radiation-interaction (ARI)
significantly influence climates at the regional and global scales (Romero et al., 2021). Assessing
the direct and indirect effects of aerosols is crucial to understanding and predicting the energy
budget and the water cycle. In the direct effect, the absorption and scattering of solar radiation by
aerosols lead to the warming of the atmosphere and cooling of the earth's surface (Zhou et al.,
2020), causing changes in the lower tropospheric stability (LTS) that further lead to modulation of
precipitating (PCs) and non-precipitation clouds (NPCs) (Andreae & Rosenfeld, 2008).
Precipitating clouds are thick clouds with significant vertical development and high moisture
content, form under unstable atmospheric conditions, such as cumulonimbus and nimbostratus,
that produce precipitation reaching the ground. In contrast,  non-precipitating clouds are typically
thin, have low moisture content, and form under stable atmospheric conditions, including cloud
types like cirrus, cirrostratus, altostratus, and stratus, which generally do not produce significant
precipitation (Houze Jr, 2014).
In the indirect effect, the water-soluble aerosols such as soil dust, sulfates, nitrates, and other
organic aerosols ejected naturally and anthropogenically serve as cloud condensation nuclei (CCN)
and ice nucleating particles (INP). Hence, aerosols affect the aerosol-cloud-interaction (ACI) by
influencing the growth of cloud droplets and cloud droplet number concentration (CDNC)
(Twomey et al., 1977; Albrecht, 1989; Jiang et al., 2002; Chen et al., 2011; Tao et al., 2012). The
increase of CDNC and decrease of cloud droplet effective radius (CER) inhibit the onset of
precipitation and increase the cloud lifetime (Albrecht, 1989). Conversely, the decrease in CDNC
and increase in CER increases the probability of precipitation rate (PR). Conversely, Stevens and
Feingold (2009) have shown that initially, more sea salt carried by high wind speed inhibits
precipitation formation. However, the same sea spray tends to seed the coalescence by producing
larger CER that leads to enhanced precipitation.
In the last few decades, most of the cultivable land of the Indo-Gangetic Plain (IGP) has been
replaced by urban developments. Due to the fastest growth of population, urbanization,
industrialization, and massive combustion of biomass and fossil fuels in residential homes and
factories, a decadal increase in aerosols is observed over IGP. The high aerosol loading may affect
the formation of tropospheric clouds and seasonal precipitation patterns (Kaskaoutis et al., 2011;
Singh et al., 2015; Thomas et al., 2021). The high aerosol loading makes IGP suitable for the study
of ACPI. Besides, frequent variations in cloud fraction (CF), extreme precipitation and drought
abrupt temperature changes (e.g., heat waves), and irregular unseasonal rains may cause major and
unavoidable hazards at local and regional levels in the future (Zhou et al., 2020).
In the last two decades, the scientific community has focused on quantification of ACI using both
observations (Feingold et al., 2003; Koren et al., 2004; Costantino et al., 2010; Wang et al., 2015;
Zhao et al., 2018, Guo et al., 2019; 2020; Anwar et al., 2022) and modeling techniques (Chen et
al., 2016, 2018; Wang et al. 2020; Zhou et al., 2020; Sharma et al., 2023). Although, a similar
recent study (Anwar et al., 2022) attempted to understand the sensitivities of ACI and the first
indirect effect of different subsets of AOD to the different conditions of RH and wind directions
and found a decrease (increase) in CER with aerosol loading Twomey effect (anti-Twomey effect)
over the monsoon (weak and moderately intensive monsoon) regions. However, the above study
excluded the other significant meteorological parameters such as LTS, PR, and $T_{850}$ and was also
limited to the monsoon regions of Pakistan only. Further, in the context of warm rain processes, it
is generally understood that the high concentration of aerosols capable of serving as CCN leads to
enhanced CDNC known as the first indirect effect (FIE) or Twomey effect (Twomey et al., 1977).
It is also widely acknowledged that CDNC plays a pivotal role in cloud microphysics and
significantly influences the onset of precipitation and retention of water in clouds called the second
indirect effect (SIE) (Gryspeerdt et al., 2016; Naud et al., 2017). Whilst, in the above study, the
analysis of CDNC is also not addressed. Therefore, the present study aims to deepen the previous
study (Anwar et al., 2022), by a long-term and detailed analysis of the ACPI including aerosol-
indirect effects for low-level liquid clouds extended over the whole IGP for understanding different
mechanisms (condensation, droplet growth and precipitation rate) of cloud and precipitation
formation. Due to the absence of in-situ measurement facilities and the constraints of limited
computational resources, the study concentrated on satellite data for specific locations across the entire IGP.
These locations were strategically chosen due to their positioning within significant aerosol belts, where
the concentration and behavior of aerosols are of particular interest. Therefore, the satellite-based approach
was chosen as it provides detailed insights into aerosol dynamics in these critical regions while also
benefiting from the broader spatial coverage of satellite data.
This study is focused on estimating the variations in sensitivities of aerosol-cloud relationship to
the variations in aerosol loading at specified meteorological conditions for low-level PCs and
NPCs in the summer and winter seasons over the IGP. This study is unique in using a large number
of samples, classification of liquid clouds in PCs and NPCs, further classification of clouds in low,
mid, and high-level clouds through joint COT-CTP histograms, quantification of the sensitivities
of FIE, SIE, total indirect effect (TIE), and ACI to CDNC. The significant meteorological
parameters considered include temperature at 850 hPa, LTS, relative humidity (RH%) at 850 hPa,
vertical velocity ($\omega$), and PR. Furthermore, by utilizing the Moderate Resolution Imaging
Spectroradiometer (MODIS) and Tropical Rainfall Measuring Mission (TRMM) data, the
correlation of cloud microphysical properties (CER and CDNC) and AOD at specified values of
LTS and cloud liquid water path (CLWP) is examined, and precipitation sensitivity at constant
macro-physical condition is estimated.

## 2. Study area and methodology

### 2.1. Study area

The selected study area (Fig. 1) comprises the upper, middle, and eastern portions of the IGP. The upper part consists of the densely populated and developed regions of the eastern part of Pakistan i.e., Karachi (24.87ºN, 67.03ºE) and Lahore (31.54ºN, 74.32ºE) whereas the middle part comprises the northern part of India i.e., Delhi (28.59ºN, 77.22ºE), Kanpur (26.51ºN, 80.23ºE), Jaipur (26.91ºN, 75.81ºE), Gandhi College (25.87ºN, 84.13ºE), Kolkata (22.57ºN, 88.36ºE), Dhaka (23.80ºN, 90.41ºE) and Patna (25.59ºN, 85.13ºE). The data analysis for the eastern part of IGP (Kolkata, Dhaka, and Patna) is documented as supplementary materials.

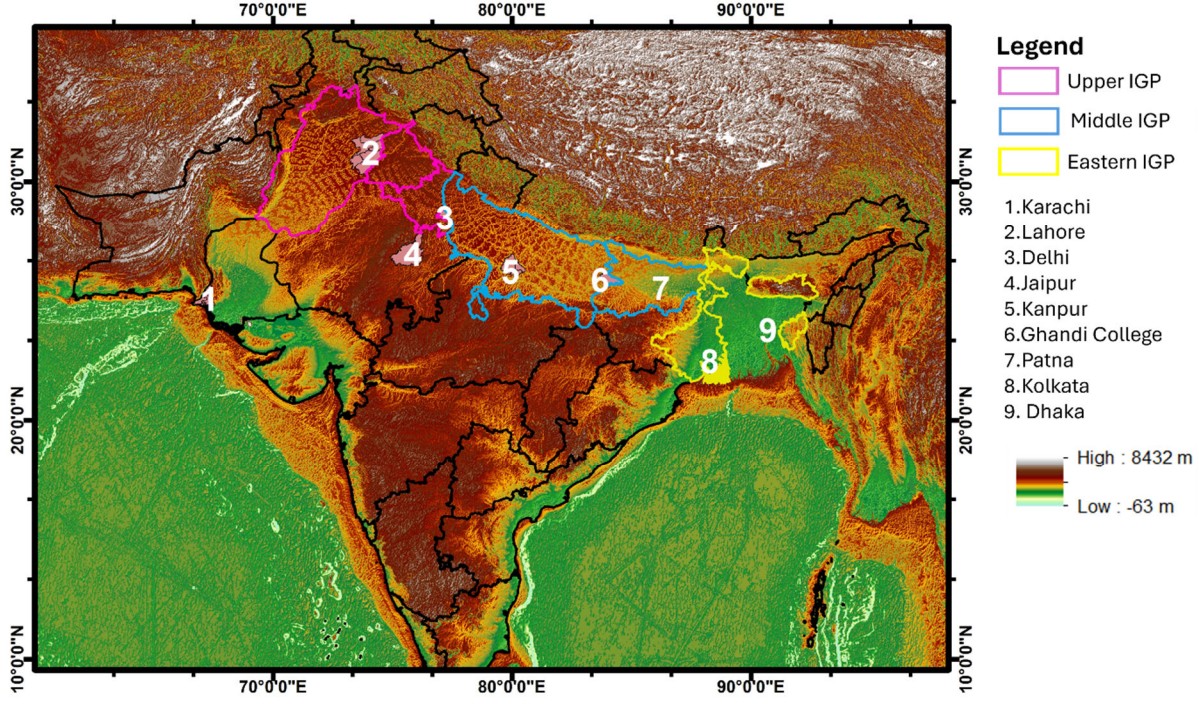

**Figure 1.** Topography of the study area.

### 2.2. Methodology

#### 2.2.1. MODIS, NCEP/NCAR reanalysis-II and TRMM data

Moderate Resolution Imaging Spectroradiometer (MODIS) is a major constituent of NASA's Earth Observing System (EOS). MODIS is orbiting with two onboard satellites, Terra and Aqua, launched in 1999 and 2002 respectively, with a range of 2330 km spanning the entire globe in a day. It provides data and information with a spatial resolution of 1° to study atmospheric processes

and physical structure (Kedia et al., 2014; Srivastava et al., 2015). This study uses the daily mean of combined dark target and deep blue AOD at 0.55 μm, cloud top pressure (CTP), cloud top temperature (CTT), CF, CER, and COT for liquid clouds from level 3 aerosol-cloud data product MOD08-TERRA. Data with AOD > 1.5 are excluded to avoid potential misidentification of aerosols as clouds. The following adiabatic approximation (Brenguier et al., 2000; Wood, 2006; Kubar et al., 2009; Michibata et al., 2014) is used to calculate CDNC (cm$^{-3}$):

$$CDNC = \left(\frac{B}{CER}\right)^3 * \sqrt[\square]{(2 * CLWP * \gamma_{eff})}$$

( 1 )

Where $B = \sqrt[3]{\left(\frac{3}{4}\pi\rho_{water}\right)} = 0.0620$, $\rho_{water}$ is the liquid water density, $\gamma_{eff}$ is the adiabatic gradient of liquid water content in the moist air column (Wood, 2006). Value of $\gamma_{eff}$ range from 1 to $2.5 \times 10^{-3}$ at a temperature of 32 K to 104 K ( Brenguier, 1991; Zhu et al., 2018; Zhou et al., 2020). The CLWP is estimated by use of

$$CLWP = \frac{5\rho_w(CER)(COT)_w}{9},$$

( 2 )

Where, $\rho_w$ is the water density at room temperature (Koike et al., 2016).

National Center for Environmental Prediction/National Center for Atmospheric Research (NCEP/NCAR) reanalysis datasets provide global reanalysis data sets that combine satellite observations with the simulation of models through data assimilation (Purdy et al., 2016). Daily data for meteorological parameters including temperature, RH%, and ω at 850 hPa are retrieved at a spatial resolution of T62 Gaussian grid (1.915° × 1.875°) from NCEP reanalysis-II datasets, and used to calculate lower tropospheric stability (LTS) defined as (Li et al., 2017):

$$LTS = \theta_{700} - \theta_{1000} \tag{3}$$

where θ is the potential temperature and the subscripts denote the pressure levels of 700 hPa and 1000 hPa.

The Tropical Rainfall Measuring Mission (TRMM) is the first Joint satellite mission between
NASA America and National Space Development Agency (NASDA) Japan, utilizing the visible
infrared and microwaves to measure the rain precipitation over tropical and subtropical regions.
The main TRMM instruments that are used to measure rain precipitation are precipitation radar
(PR) and  TRMM Microwave Imager (TMI). Where PR is operating at a frequency of 13.8 GHz
and TMI is a passive microwave radiometer consisting of nine channels. A calibrated data set
TRMM-2B31 of TRMM Combined Instrument (TCI) for TRMM Multi-Satellite Precipitation
Analysis (TMPA) is formed from an algorithm that uses TMI and PR. The product TMPA 3B42
gives the rain precipitation averages on a daily and sub-daily basis. In the current study, the data
product TMPA or TRMM 3B42 is used for the retrieval of PR daily. The spatial resolution of
TRMM 3B42 is 0.25º × 0.25º  and is available from the year 1998 to till date.
2.2.2.  Methodology
The present study is designed to analyze and quantify the ACPI for PCs and NPCs in winter and
summer under a variety of meteorological conditions. The daily mean data of each parameter for
warm clouds are retrieved from the respective satellites and NCEP/NCAR reanalysis-II for each
study site. Subsequently, the VLOOKUP function in Microsoft Excel is applied to filtering out
counts where data is not available, searching for values of a parameter in the first column, and
retrieving the values of other parameters in the same rows on the corresponding dates in a large
dataset. The data are then segregated into two subsets for the summer and winter seasons. Based
on precipitation data from TRMM, the subsets are further divided into precipitating and non-
precipitating clouds.
The sensitivities of cloud parameters to CDNC are analyzed through the following formulation
considered from previous studies (Zhou et al., 2020):
$$\frac{dln(COT)}{dln(CDNC)} = -\frac{dln(CER)}{dln(CDNC)} + \frac{dln(CLWP)}{dln(CDNC)} \qquad (4)$$
In this study, the term on the left side of equation (3) is defined as total indirect aerosol effect
(TIE), and the first and second terms on the right side of the equation are defined as the first indirect
aerosol effect (FIE), *and second indirect effect* ($SIE$), *respectively*. Similarly, the sensitivity
of CDNC to AOD is evaluated by employing the index of ACI:

$$\text{ACI}_{\text{CDNC}} = \frac{dln(CDNC)}{dln(AOD)} \qquad (5)$$

The sensitivity of PR to CDNC is calculated from the following equation (Jung et al., 2012) :

$$S_0 = \left(-\frac{\partial ln(PR)}{\partial ln\,(CDNC)}\right)_{COT} \qquad (6)$$

## 3. Results and Discussion

### 3.1. Regional and seasonal distribution of AOD

AOD is a commonly used proxy for aerosol concentration in the atmosphere and is analyzed here (Fig. 2-3).

IGP characteristically exhibits a diverse and massive pool of aerosols due to its unique topography. The western part of IGP is a coastal location and inlet for the westerly winds. Therefore, dry regions and the Arabian Sea in the west contribute dust, sea salt, and water vapors to the region. The Himalayas in the north act as barriers to the winds, leading to the trapping of aerosols over the central part of IGP. Therefore, this region exhibits a high concentration of anthropogenic aerosols. The Bay of Bengal in the east allows southeasterly winds to enter passing across Dhaka, Kolkata, and Patna to Delhi and Lahore (Hassan et al., 2002; Anwar et al., 2022). The westerly and easterly winds traverse forested hilly terrain, rivers, and lakes elevating humidity levels and initiating the cloud formation by activation of the newly originated small aerosol particles as CCNs and cloud formation affecting the local microclimate.

Fig. 2 shows a decadal variation in time average maps for combined dark target and deep blue AOD retrieved at 0.55 µm over the entire study area for the years (2001-2010) and (2011-2021). Also, Table 1 illustrates the percentage change in decadal averaged values of AOD. The results indicate that AOD exhibits a decrease over Karachi (-1.9%) and Jaipur (-0.5%). An increase in AOD is observed over Lahore (5.2%), Delhi (9%), Kanpur (10.7%) and Gandhi College (22.7%). Similarly, Table 1S shows the decadal change in AOD over Kolkata (18%), Dhaka (22.6%), and Patna (23.3%). Similar to Gandhi College, an increase is observed over all three areas. Reasons for the increase of aerosols include multiple sources of aerosols, human behavior, socio-economic

development at local and regional levels, and unique topography for the persistence and retaining of aerosols.

Fig.3(a-b) shows the probability density function (PDF) for AOD, illustrating different distributions in the summer and winter seasons. Fig.3a shows that the distribution of AOD over Delhi, Kanpur, and Gandhi College is similar. However, a shift in the peak value of PDF towards high values of AOD over Lahore and low values over Jaipur illustrate comparatively high and low aerosol concentration in the summer season over Lahore and Jaipur respectively. Likewise, Fig. 1S shows the seasonal PDF values of AOD over Kolkata, Dhaka, and Patna. The results indicate similar seasonal distribution functions over all three areas of eastern IGP. In both seasons PDF peaks for high values of AOD are observed over Patna showing a high concentration of aerosols as compared to Kolkata and Dhaka.

The loading of high concentrations of aerosols is owing to the high density of population, industrialization, and human activities. The major sources of aerosols in all months of the year include vehicular emission originating from old transport facilities, emission of smoke and soot during consumption of biomass for cooking, heavy industrial emission, and aerosols produced in seasonal harvesting and crop residue burning. All these sources produce organic aerosols which are characterized as hydrophilic particles and have the potential to act as CCN. Likewise, the soil dust particles also act as good CCN due to their hydroscopic nature (Sun & Ariya, 2006). Moreover, the meteorological conditions also play a substantial role in enhancing AOD values such as the uplifting of loose soil dust and swelling of aerosols due to holding the water vapors (wv) for a long time (Masmoudi et al., 2003; Alam et al., 2010; Alam et al., 2011;). Also, the lower but flat PDF curve demonstrates low values of AOD over Karachi. Ali et al., 2020 associated the low AOD values over Karachi with the westerly and southwesterly wind currents at tropospheric level. However, the decreasing trend in AOD over the coastal city may also be attributed to the variations in other meteorological parameters like T and RH.

As compared to the summer season, the pattern of PDF in winter is significantly different as shown in Fig. 3b. The low value of PDF (0.5) for the high value of AOD (0.9) over Karachi illustrates a comparatively pristine atmosphere. Similarly, the PDF peaks for Lahore, Delhi, and Jaipur (0.7, 0.7, and 0.8) indicate comparatively high AOD over Delhi. Likewise, the distribution over Kanpur and Gandhi College similarly illustrates similar values of AOD (1.1 and 1.2 respectively). These

high values of AOD are attributed to the high emission of anthropogenic aerosols at local and
regional levels over the central part of IGP (Delhi, Jaipur, Kanpur and Gandhi College).
Few authors attributed the reduced values of AOD in the winter season to the wet scavenging and
suppressed emission of aerosols from the earth surface (Alam et al., 2010; Zeb et al., 2019).
However, in our case, the low (high) values in winter (summer) are associated with the dispersion
of fine (course) mode particles due to the variations in meteorological conditions.

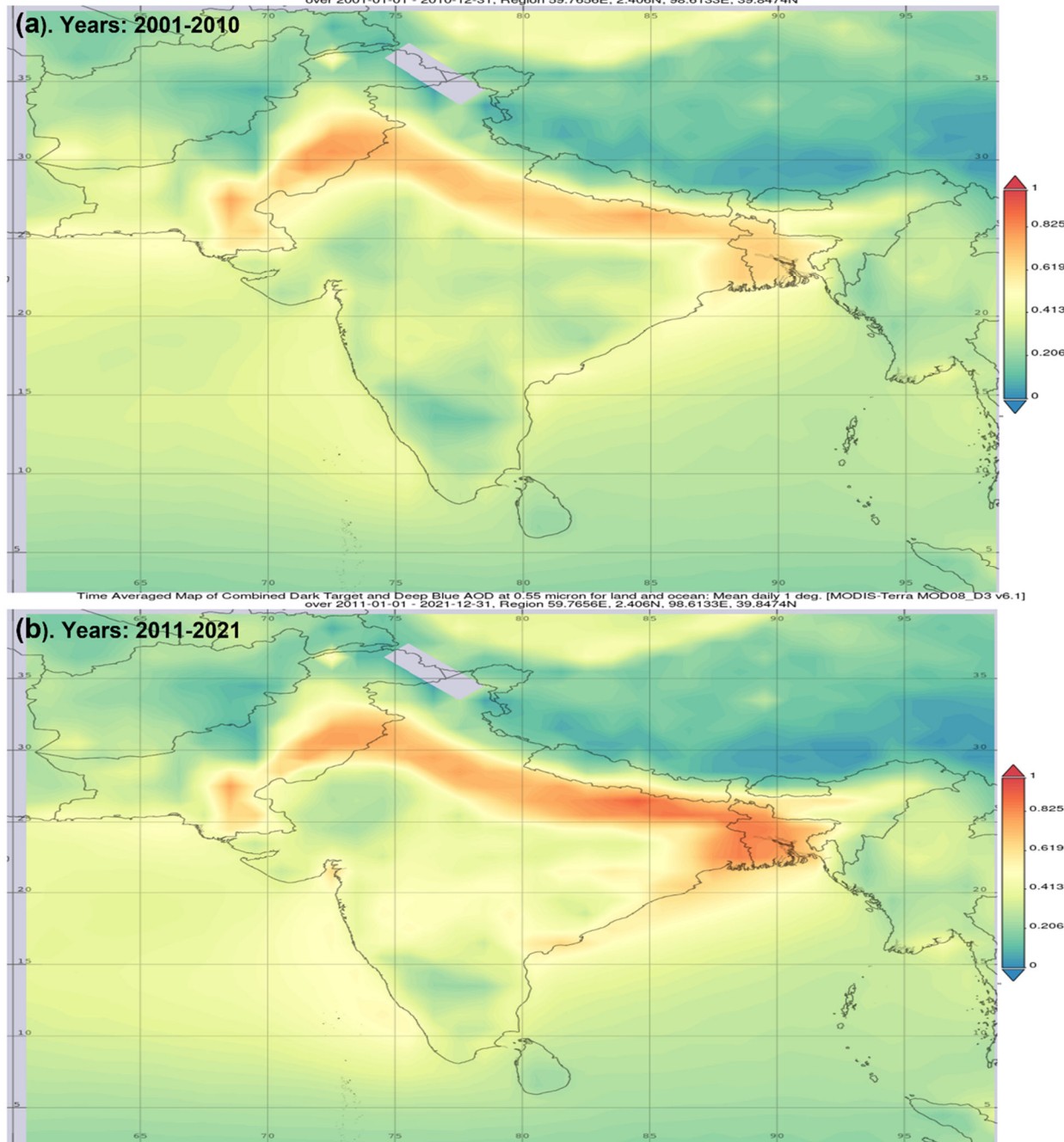

**Figure 2.** Decadal increase (year: 2001-2010 and 2011-2021) in AOD over study sites.







**Table 1.** Decadal percentage variations in average values of AOD over all study areas

|  | Karachi | Lahore | Delhi | Kanpur | Jaipur | Gandhi College |
|---|---|---|---|---|---|---|
| Total number of counts | 5902 | 6171 | 5823 | 5201 | 5907 | 5125 |
| Decadal change in AOD | -1.9% | 5.2% | 9% | 10.7% | -0.5% | 22.7% |


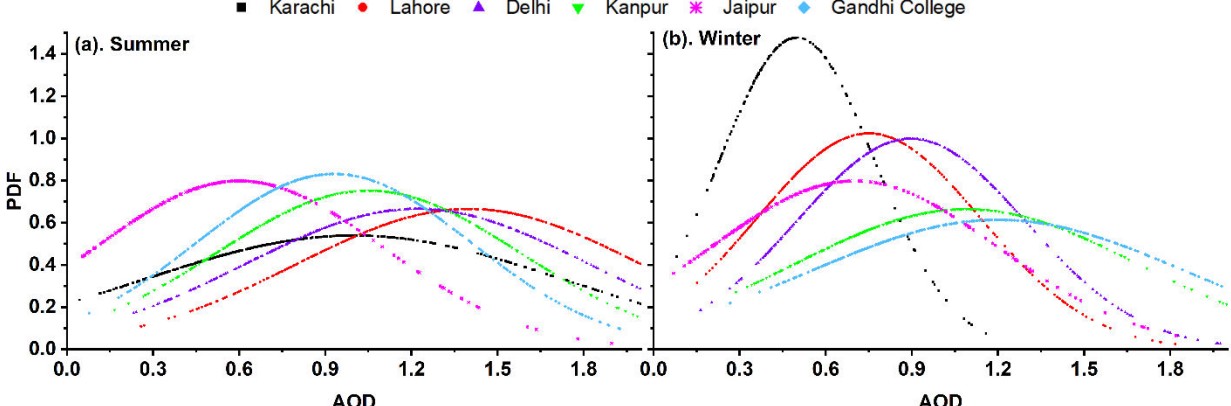

**Figure 3.** The probability density function (PDF) of AOD over study sites is shown (a) and (b) for the summer and
winter seasons respectively.
3.2. Climatology of meteorological parameters
Generally, LTS has relationships to factors such as temperature, humidity, wind patterns, and
atmospheric pressure over extended periods. It is also widely acknowledged that atmospheric
stability, temperature, RH wind speed, and direction play a significant role in cloud formation
(Yang et al., 2015; Tao et al., 2012). Therefore, the influence of long-term variations in the said
meteorological parameters is considered in the current study. The variations in meteorological
parameters have an unavoidable impact on ACPI. The parameters considered in this study include
the temperature, LTS to determine the lower atmospheric stability and instability that influence the
process of cloud and precipitation formation through its significant implications on evaporation
and convection of the air parcel, the RH% to estimate the level of wv and the ω to assess the
suitable atmospheric dynamics. Fig.4 shows the variations in LTS values for NPCs and PCs in the
winter and summer seasons. In the winter season, the LTS values are high for NPCs and
comparatively lower for PCs over entire study areas. In the summer season, the scenario is reversed
with high values for PCs but low values for NPCs, suggesting a stable tropospheric layer on rainy

days. This stabilization may be attributed to the cold pools generated by the evaporation of falling rain droplets (Wu et al., 2017).  The lower LTS values for NPCs in the summer season suggest the likelihood of stronger instability that causes a high potential for vertical motion and the development of thunderstorms. However, Karachi exhibits a distinct pattern of LTS with the highest values in each case, which indicates the existence of the most stable tropospheric layer in Karachi due likely to moist and cold sea breeze due to the city's coastal location.

The median values computed for the remaining meteorological parameters considered in this study are listed in Table 2. The high values in each case are indicated in bold and the low values are italicized. The results show that in the winter season, the temperature at 850 hPa ($T_{850}$) is relatively high for NPCs ranging from 281 K to 285.6 K. The increase in RH% for PCs during winter ranged from (59.5)% to (71.5)%. Also, the $\omega > 0$ for NPCs and $< 0$ for PCs in the winter season.

In the summer season, it is observed that $T_{850}$ is comparatively higher than that for the winter clouds and ranges from 298.3 to 300.2 K and 296.5 to 298.3 K for NPCs and PCs respectively. The high values of $T_{850}$ are due to intense solar fluxes in the summer season that keep the temperature of the earth's surface and adjacent atmospheric layer higher.  Also, the increase in RH% during summer ranged between 33.5-51.7 % for NPCs. The reason for the high values of wv and RH% is mainly the suitable thermodynamical conditions such as evaporation and convection due to the high temperature of the earth's surface and air (Sherwood et al., 2010). The results show high values of RH% 70.1% (85%) in the winter (summer) season for PCs over Gandhi College. Conversely, notable fluctuations in RH% are observed over the coastal city, of Karachi, with values of 71.5% (65.9%) in winter (summer). Similarly, Fig. 2S and Table 2S show the LTS conditions for PCs and NPCs. The high LTS values indicate more stable conditions over Dhaka. Similarly, Table 2S shows the seasonal average values for other meteorological parameters. The results indicate high values of $T_{850}$, RH%, and $\omega$ 295.5 (297.5) K, 88.8 (83.5)%, and -0.19 (-0.17) m/s respectively for PCs (NPCs) for over Patna in summer.

Besides, during the last two decades, the wv and fog over the Arabian Sea increased (Verma et al., 2022). Therefore, the high values of wv and RH% in summer months are due to the high-speed zonal winds that blow in the summer season and transport water vapors and sea salt from the surface of the Arabian Sea and hydrophilic aerosols such as soil dust from deserts of Iran, Pakistan, and India to IGP.  Moreover, during the winter season, elevated humidity levels are noticeable over

IGP, particularly in the vicinity of Gandhi College. This increased humidity is a result of evapotranspiration driven by agricultural practices, irrigation, the presence of rivers and lakes, and the introduction of moist, cold air from western winds (Nair et al., 2020). Where $\omega < 0$ for PCs over all study areas except Karachi.

The distinct variations in meteorological parameters reveal the occurrence of clouds with diverse properties. The detailed analysis of such clouds is given in the next subsections.

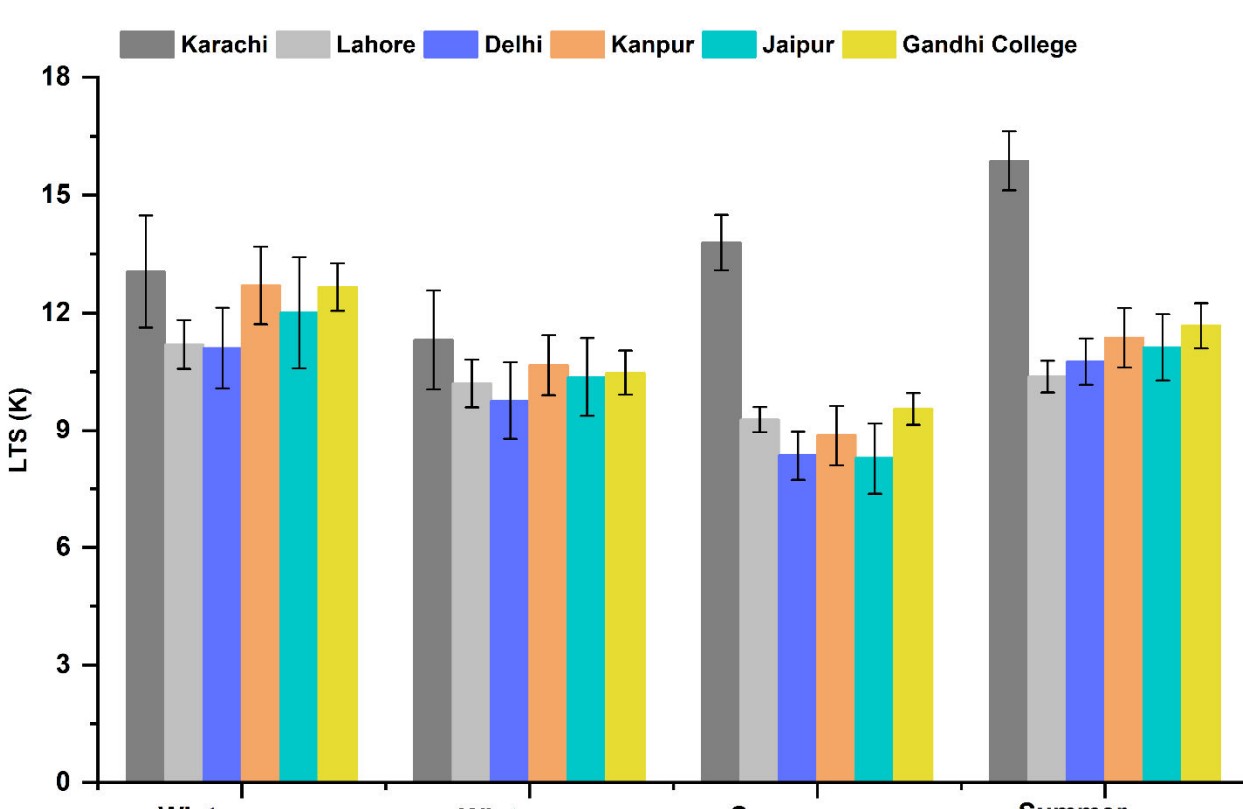

**Figure 4.** Variations in lower tropospheric stability (LTS) over all study sites for PCs and NPCs in winter and summer seasons, the error bars show the standard deviation (SD) values.

**Table 2.** Median values of meteorological parameters for PCs(NPCs) in summer and winter seasons. Maximum values are for both types of
clouds shown in bold and minimum values are indicated in italics.

| | Winter Season | | | Summer Season | | |
|---|---|---|---|---|---|---|
| | $T_{850}$ (K) | RH% | $\omega$ (m/s) | $T_{850}$ (K) | RH% | $\omega$ (m/s) |
| Karachi | **284.6 (285.8)** | **71.5** (38) | -0.038 (*0.030*) | *295.9* (298.8) | 65.9 (45.9) | *0.005* (-0.003) |
| Lahore | *280.5* (*281.2*) | *59.5* (35.5) | *-0.02* (**0.065**) | **298.3 (300.2)** | 65 (*33.5*) | -0.028 (*0.025*) |
| Delhi | 284.2 (283.1) | 60.2 (*33.8*) | **-0.1** (0.04) | 296.5 (299.4) | 64.2 (42) | *-0.05 (-0.001)* |
| Kanpur | 283.8 (284.1) | 65.7 (36) | **-0.1** (0.048) | 296.5 (298.4) | 73.7 (43.6) | *-0.13* (-0.08) |
| Jaipur | 283.9 (284.1) | 66 (40.5) | -0.065 (0.049) | 296.8 (298.7) | *64* (**51.7**) | *-0.04 (-0.029)* |
| Gandhi College | 283.2 (284.1) | 70.1 (**45.7**) | **-0.1** (0.05) | 296.9 (*298.3*) | **85** (42.5) | ***-0.16 (-0.11)*** |






### 3.3. Regional and seasonal distribution of clouds and precipitations

    3.3.1. Regional and seasonal differences in cloud occurrence and its microphysical structure

Fig.5 shows the frequency of occurrence of precipitable clouds and the total number of cloudy days. Chen et al. (2018) suggested the COT to be an effective measure for assessing the clouds and potential for precipitation. In our case, to avoid any overestimation, the COT data are aligned with PR data on corresponding dates and then filtered to include COT ~ > 5 for PCs. The results show that in the winter season, the frequency of clouds is low over Karachi and high over Lahore and Gandhi College. The results suggest the high number of PCs only over Lahore. In the summer season, a high number i.e., 74 % of the total data counts over Gandhi College are identified as cloudy days, 60 % of which are PCs. Similarly, most of the clouds over Lahore, Delhi, and Jaipur are PCs. Conversely, the least number of PCs (6 %) are found over Karachi. Likewise, Fig. 3S shows the total number of cloudy days and the number of days on which PCs occurred. The high occurrence of clouds is observed over Kolkata 83% (60%) and Dhaka 91% (69%) in the summer (winter) season. The high occurrence of PCs in summer is due likely to the significant impact of elevated aerosols with the southwesterly winds on the summer monsoons and the occurrence of PCs. Therefore, Kolkata and Dhaka are of critical importance from the perspective of aerosol loading and ACI (Dahal et al., 2022).

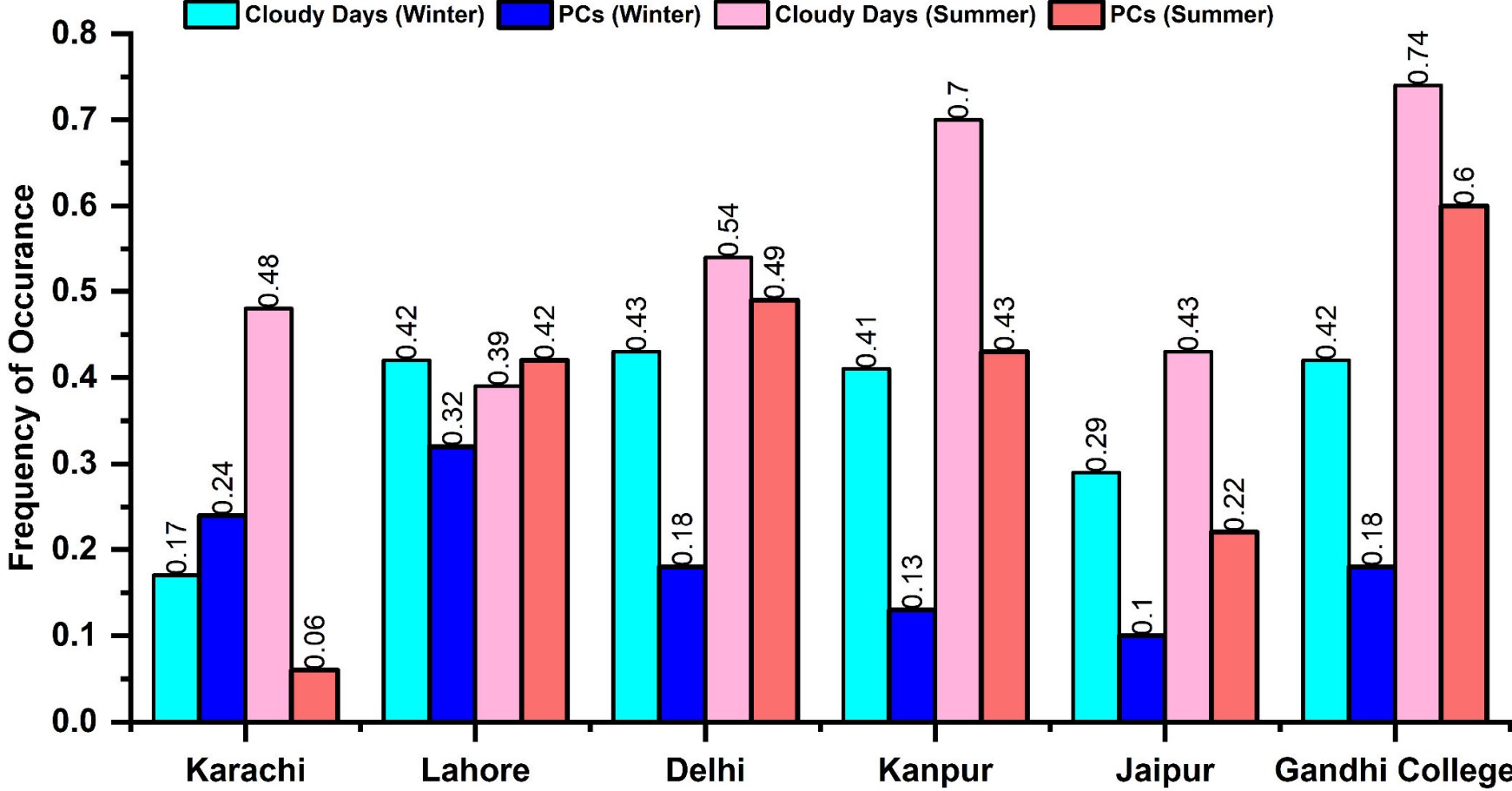

**Figure 5.** The frequency of occurrence of total cloudy days (including PCs and NPCs) and only PCs is shown for both winter and summer seasons.

Table 3 shows the criteria adopted from previous papers (Rossow & Schiffer, 1999; Wyant et
al., 2006; Sharma et al., 2023) for further classification of NPCs and PCs into different types
of clouds. The aim of identifying the cloud types is to assess the cloud regimes and their vertical
structure for a better understanding of ACPI. Following table 3, Fig. 6 shows joint histograms
of COT-CTP displaying the median values of CF for nine different types of clouds. For a quick
visual comparison, the cloud types are ordered from low to high-level clouds. Also, for each
histogram, the bins of COT and CTP are located on the x- and y-axis respectively. The CF of
each bin is represented with the colored bar with its value mentioned in the histograms as shown
in Fig. 6.
The results exhibit noticeable differences in the pattern of cloud regimes over all study areas.
The diverse CF values are observed in the winter and summer seasons for NPCs and PCs over
Karachi. In the winter season, only stratus NPCs (23 < COT <60, 800 > CTP > 680 hPa) are
dominant with CF ~ 0.9. While, in summers, the high value of CF ~ 0.9  for low and
intermediate thickness of high-level clouds such as Cirr-Stratus NPCs (3.6 < COT < 23, 180<
CTP < 440 hPa) are observed. Similarly, the types of PCs in both summer and winter seasons
that occurred with CF ~1.0 include cirrus and cirrus-stratus. The relatively reduced value of CF
for thick NPCs in winter and PCs in summer is attributed to the low values of AOD and high
values of LTS. The results depicted slight differences and similarities in CF values for thick
and thin NPCs respectively in the winter season for all areas except Karachi. Besides, the high-
level PCs are identified in the two bins of CTP (180< CTP < 440 hPa) and  (440< CTP < 680
hPa) over all study areas. The formation of these similar types of PCs in winter is associated
with the similarities in ω, LTS values, and aerosol concentration.
Likewise, in the summer season, the matrices of PCs and NPCs exhibit a wide range of cloud
types. However, the CF values are comparatively high for PCs. Most of the identified PCs are
formed in the two bins of CTP (180< CTP < 440 hPa) and (440< CTP < 680 hPa) with CF
values ranging from 0.8 to 1.0. The results suggest low values of CF  for the low-lying thick
NPCs over all study areas. Moreover, the results illustrate a more frequent occurrence of all
three types of thick NPCs in one bin of COT (23 < COT< 60) and all the three types of high-
level NPCs for CTP (180 < CTP < 440 hPa) over Delhi, Kanpur, and Gandhi College.
Therefore, these are considered the cloudiest regimes. Besides, contrasting regional variations
are also observed in PCs. The maximum CF values for all types of PCs are observed over
Kanpur and Gandhi College. Similarly, relatively good values of CF in a bin of  COT (23 <
COT< 60) and a bin of CTP (180 < CTP < 440 hPa) over Lahore, Delhi, and Jaipur depict the
frequent occurrence of thick and high-level PCs respectively. In addition, among all the
estimated low-level PCs, cumulus and strato-cumulus exhibit good CF values (0.7) over
Kanpur and Gandhi College. The formation of thick clouds can be attributed to the enhanced
convection process due to atmospheric instability.
**Table 3.** Classification of clouds based on CTP – COT joint histograms.

| CTP (hPa) | COT | | |
|---|---|---|---|
| | 0-3.6 | 3.6-23 | 23 to >60 |
| 440 to <180 | Cirrus | Cirr-Stratus | Deep convection |
| 680-440 | Alto-Cumulus | Alto-Stratus | Nimbo-Stratus |
| <800 to 680 | Cumulus | Strato-Cumulus | Stratus |


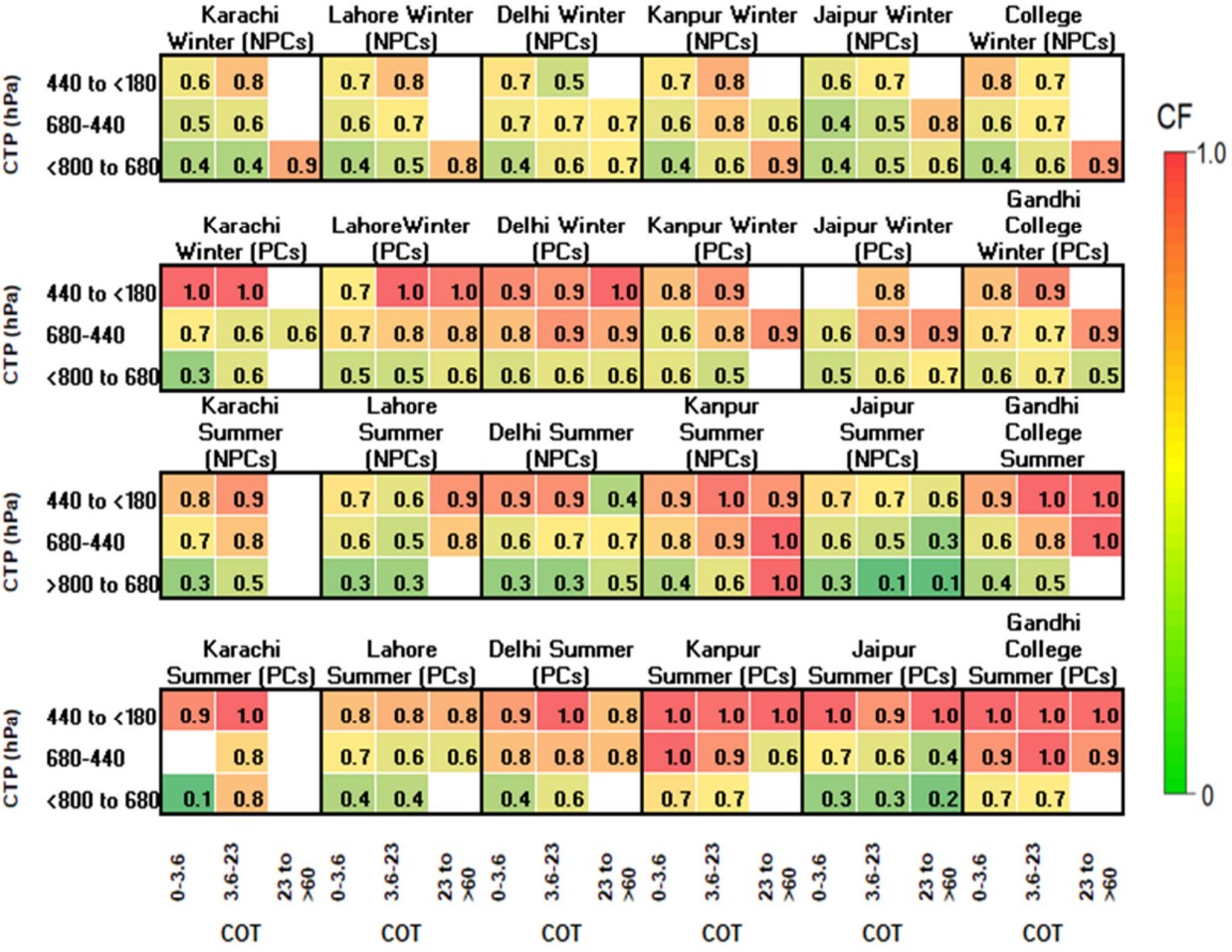


**Figure 6.** Types of NPCs and PCs in winter and summer season

After estimating the cloud types, Fig. 7 shows the probability distribution function (PDF) of
cloud microphysical properties for the identification of differences in the microstructure of
NPCs and PCs in the summer and winter seasons. From the results, it is depicted an
approximately similar pattern for the CER of NPCs in winter. However. the clouds have high
peaks of PDF for lower values of CDNC over Karachi. The low number of CDNC results in
thin NPCs as shown in Fig.7. Similarly,  Fig. 7(c and g) shows the microstructure of NPCs in
summer. The results indicate that as compared to CER values in winter, the probability of CER
$>\sim$15 µm is high in the summer season. However, the high peak for CER < 15 µm is observed
over Karachi. Similarly, the CDNC shows a high probability for CDNC > 50 cm$^{-3}$ with high
PDF values over Karachi. Where the lowest number of CDNC is observed over Lahore
indicating the formation of high-level thin NPCs in summer.
Fig. 7(b and f) shows the distribution pattern of CER and CDNC of PCs in the winter season.
It is observed that the distribution of CER for PCs is like that for NPCs in the winter season.
However, PDFs have peak values for relatively higher CDNC, which illustrates the occurrence
of thick clouds. Fig. 7(d and h) shows the variations in CER and CDNC in the summer season.
The results show a wider distribution for CER > ~15 µm and higher peaks for CDNC > ~ 50
cm$^{-3}$ suggesting the formation of thick PCs in summer as shown in Fig.6.


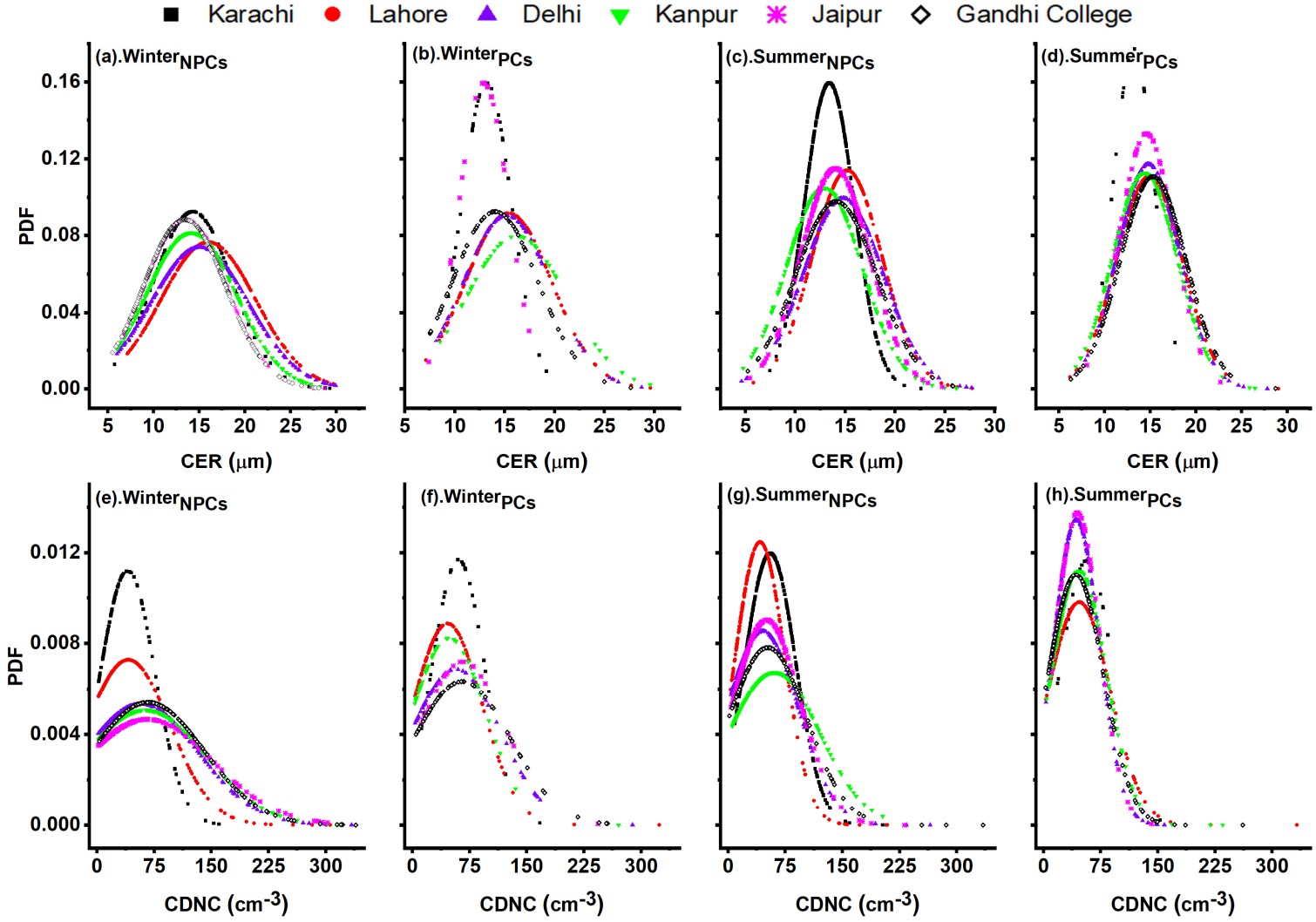

**Figure 7.** Probability density function (PDF) of precipitating (PCs) and non-precipitating clouds (NPCs) in the winter and summer season

## 3.4. Aerosol-Cloud-Precipitation Interaction (ACPI)

In the following sections, ACPI is analyzed and discussed in detail for PCs and NPCs in the summer and winter seasons.

### 3.4.1. Aerosol effects on cloud properties

The impact of aerosols on CDNC and CER of PCs and NPCs is illustrated as scatter plots in Fig. 8-9. The quantification of the AOD-CER and AOD-CDNC relationships is demonstrated through detailed linear regressed slopes, regression coefficients ($R^2$), and Pearson's correlation coefficient (R). The color bar represents the variations in LTS. The results show that the two-sample student's t-test is carried out to analyze the AOD-CER and AOD-CDNC relationship in view of statistics. The results illustrate that the relationships are statistically significant at a 95% ($p < 0.05$) significance level for all study areas. Fig.8 shows that in the winter season, the AOD-CER correlation is good for PCs and weak for NPCs. The results also show that the LTS values are higher for NPCs. The weak AOD-CER correlation may be linked to the inhibition of droplet growth due to less soluble aerosols, originating from biomass burning (Kang et al., 2015). In our case, all the selected study areas are among the most urbanized and industrialized areas of IGP. Therefore, most of the prevailing aerosols are the less soluble soot and BC particles. That weakened activation of cloud droplets inhibits the formation of PCs and evaporates to higher altitudes thereby increasing the droplet residence time (Kumar & Physics, 2013). Besides, the results show a contrasting pattern of LTS values. Although RH over Karachi (38.3±9 %) is higher than over the other study areas (shown in Table 2), the negative AOD-CER correlation is observed over Karachi due to its coastal location, the low value of AOD and high level of LTS.

Fig. 9 illustrates the AOD-CER and AOD-CDNC correlation in the summer season. The results depict a more significant and positive AOD-CER correlation in the summer season than winter season. Unlike the winter season, high LTS values are observed for PCs. Yuan( 2008) associated the positive AOD-CER correlation with the soluble organic aerosols. Myhre et al. (2007) hypothesized that the positive AOD-CER correlation is a maximum for low CTP and a minimum for high CTP. Hence, in our study, referring to the approximated CF values shown in Fig.6, the significant and positive AOD-CER correlation under unstable atmospheric conditions resulted in thick and high-level clouds. Furthermore, it is observed that CER and CDNC values for NPCs increase with increasing instability. Meanwhile, the enhanced process of droplet activation may result in large AOD, higher CER, giant, and fewer CCN (Yuan, 2008).

Therefore, the weak correlation of AOD with CER and CDNC may be due to the
anthropogenically ejected water-soluble organic aerosols and a smaller number of CCN.
Fig. 5S and 6S show the impact of AOD on CER and CDNC for PCs and NPCs in winter and
summer respectively. The results indicate a positive and weak AOD-CER correlation of 0.2,
0.07, and 0.004 for NPCs over Kolkata, Dhaka, and Patna respectively, and for PCs (0.08) over
both Kolkata and Patna. Similarly, a positive and weak AOD-CDNC is observed over all areas
for PCs. Likewise, Fig. 6S also illustrates weak AOD-CER correlation is 0.06, 0.2, and 0.12
for both types of clouds in summer. As compared to other areas, the correlation analysis is less
significant over Karachi, Kolkata, Dhaka, and Patna. This can be attributed to the persistence
of diverse aerosol types influenced by their coastal locations, different meteorology, and the
alternating inflow and outflow of easterly and westerly winds.
Recent advances in remote sensing led to cost-effective solutions and an increase in available
data at various temporal and spatial resolutions to bridge scientific gaps among different
disciplines. While satellite-based retrievals have many advantages over in-situ and ground-
based measurement such as broader regional coverage and enhanced spatial resolution, they
are still prone to considerable uncertainties owing to the indirect nature of remote-sensing,
retrieval algorithms, thermal radiance, infrequency of satellite overpasses, and cloud top
reflectance (Hong et al., 2006; Tian et al., 2010; Hossain et al., 2006). In our study, apart from
the aforementioned factors contributing to the uncertainty, any residual cloud contamination
could also lead to biased retrieval of AOD. Likewise, satellite-based retrievals for cloud
properties are crucial to understanding the pivotal role of clouds in climate and the role of
clouds is still a dominant source of uncertainty in the prediction of climate change. These,
uncertainties in AOD and retrievals of cloud properties also propagate through the modeling
process, potentially leading to less accurate climate predictions. Likewise, these uncertainties
appeared to influence the findings in the current investigation. For instance, a limited
correlation between AOD and CER is observed over Lahore, particularly in cloudier regimes
as depicted in Fig. (5-6). This contrasts with robust impacts documented in earlier studies
(Michibata et al., 2014). However, high sensitivity of SIE is observed for PCs particularly in
the winter season indicating the delay in the onset of precipitation and more retention of clouds.

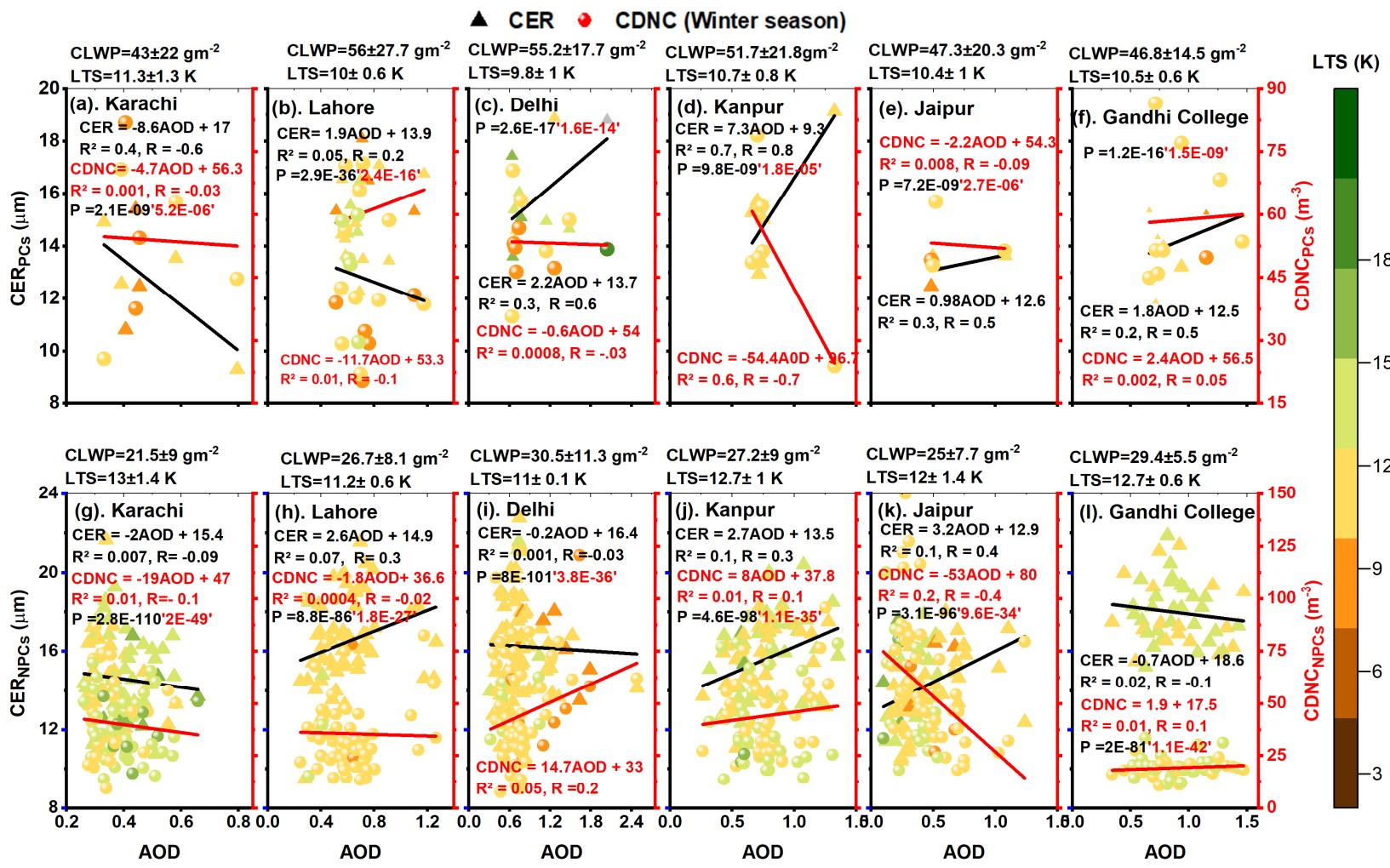

**Figure 8.** AOD-CER and AOD-CDNC regression and correlation coefficient considered at 95% confidence level for PCs and NPCs over all study areas in the winter season.




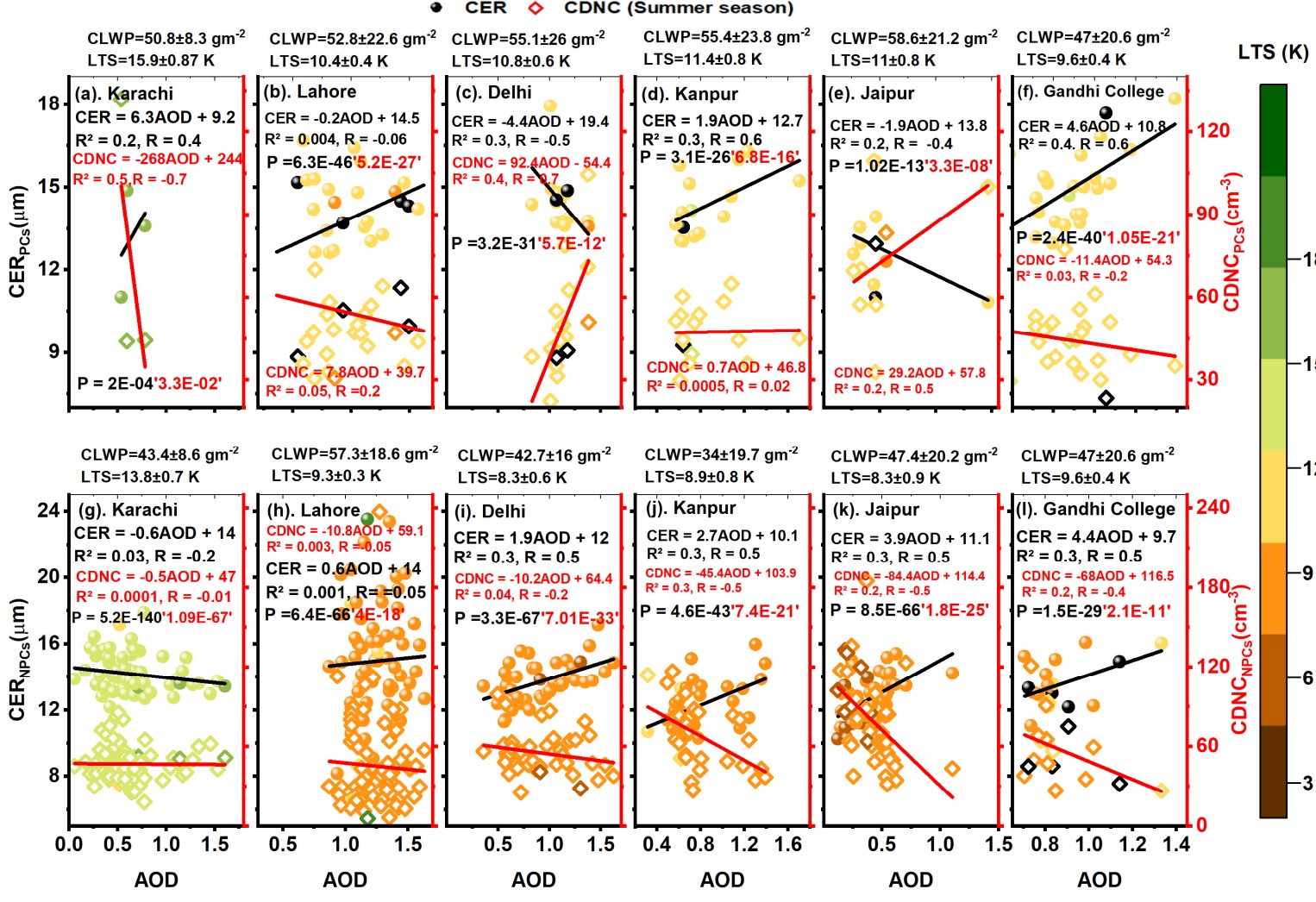

**Figure 9.** Same as Fig. 8 but in the summer season.

*3.4.2.   Seasonal variations in sensitivities of aerosol-cloud indirect effects and ACI*
Fig.10 shows an assessment of four ACI sensitivities in terms of CDNC using daily mean
values of MODIS observations available over the entire study area. Studying the effects of
aerosols on the co-located clouds is a challenging task due to the overestimation of thin clouds
in AOD retrievals. Therefore, to minimize the propagation of AOD retrieval errors in ACI, the
current study attempted to estimate the sensitivities of different cloud mechanisms to CDNC.
The sensitivity of CER to CDNC is assessed as a signature of FIE as shown in Fig.10a. The
positive values illustrate that CER decreases with an increase in CDNC revealing the
occurrence of the Twomey effect. While the negative values depict the anti-Twomey effect.
Tripathi et al., (2007) divided IGP into four regions western, central, eastern part of IGP, and
the foothills of the Himalayas. Their results depicted a high concentration of dust in the western
part, and an increase in anthropogenic aerosols as one moves from the western to the eastern
part of IGP. Therefore, they attributed the resulting strong indirect effect in winter to the high
concentration of regional anthropogenic pollution. However, in our case, the FIE is investigated
for both PCs and NPCs in both seasons. The resulting approximations in the winter season
show strong (weak) sensitivity of FIE for PCs (NPCs). Similarly, the estimated sensitivity of
FIE for all NPCs and PCs is also positive in the summer season. Fig. 7S(a) shows sensitivities
for FIE in both seasons for PCs and NPCs. The results indicate high values of sensitivity FIE
in the winter season which is similar to the results for Karachi, Lahore, Delhi, and Kanpur as
shown in Fig. 10 a. This is attributed to high levels of aerosol emission from residential heating
and industrial activities. Furthermore, the results illustrate higher values of FIE in summer. This
is attributed to the massive aerosol loading due to aerosol carried by winds and originating
from anthropogenic activities and unstable meteorology.
Fig. 10b illustrates the sensitivity of CLWP  to CDNC as a proxy for the evaluation of the SIE
or lifetime effect. The positive sensitivity estimated for all NPCs and PCs suggested that the
CLWP increases with an increase in aerosol. Further, the results show that the sensitivity of
SIE is stronger for PCs in winter which indicates the delay in the onset of high PR. Similarly,
the results show that the SIE sensitivity values are higher for PCs than for NPCs in the
corresponding seasons. Therefore, the results depict that the lifetime of PCs is greater than
NPCs. This is attributed to the high level of RH for PCs as shown in Table 2.  Fig. 10 (a and b)
shows that the FIE sensitivities are weaker than SIE.
Fig. 10c shows the TIE in terms of the sensitivity of COT to CDNC.  The results illustrate
positive values of sensitivity for all NPCs and PCs which indicate that COT increases with an
increase in aerosol concentration. The results also reveal that the sensitivity of TIE is a linear
sum of the sensitivities of FIE and SIE. Further, the results also suggest that the variations in
TIE sensitivity are largely dependent on SIE.
Fig.10d shows the sensitivity of CDNC to AOD as an estimation of ACI in terms of CDNC.
The positive values show the increase in CDNC with the increase in AOD. Therefore, positive
ACI reflects the inhibition of precipitation formation. Whilst, the negative values illustrate the
decrease in CDNC and enhanced PR (Fan et al., 2018). The results depicted relatively large
and positive sensitivities for NPCs in winter over Lahore, Delhi, and Kanpur, which inhibits
the onset of rainfall. The Sensitivity of ACI for NPCs in summer is positive over Karachi and
Lahore and negative over Delhi, Kanpur, Jaipur, and Gandhi College. Ackerman et al. (2004)
associated the negative $ACI_{CDNC}$ with the wet scavenging and mixing of air by entertainment.
In our case, negative ACI may be due to the growth of CER and a decrease in CDNC with
aerosol loading under unstable conditions(shown in Fig. 9). Further, the magnitude of
sensitivity for PCs in summer is low. This can be due to the droplet growth through collision
coalescence and wet scavenging in thick clouds, decreased dependency on CCN.


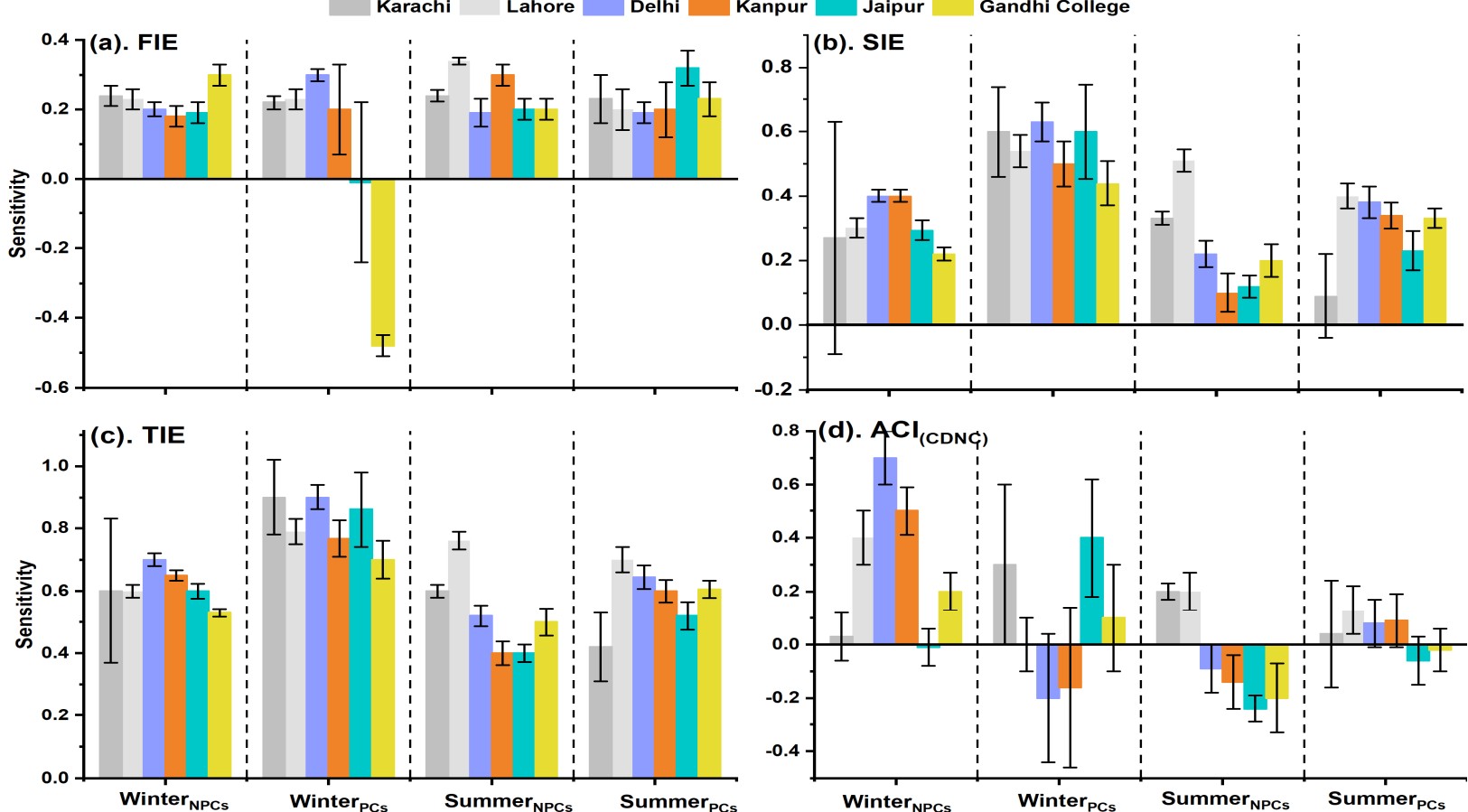

**Fig. 10.** The sensitivity matrices estimated for an aerosol-cloud relationship using CDNC are shown in (a) FIE = $-\left(\frac{\partial ln\ (CER)}{\partial ln\ (CDNC)}\right)$ (b) SIE = $\left(\frac{\partial ln\ (cLWP)}{\partial ln\ (CDNC)}\right)$, (c) TIE = $\left(\frac{\partial ln\ (COT)}{\partial ln\ (CDNC)}\right)$ and (d) ACI = $\left(\frac{\partial ln\ (CDNC)}{\partial ln\ (AOD)}\right)$. Where the error bars show the standard deviation (SD).

*3.4.3.  Aerosol effects on precipitation*
Fig. 11 shows the average values of PR in mm/day retrieved from TRMM. The results show
an obvious seasonal difference in precipitation occurrence. The reason for the high (low) PR
values is due to the suitable meteorological conditions including high (low) LTS values for PCs
in the summer (winter) season (shown in Fig. 8-9). The stable atmospheric condition with a
high LTS value in winter serves to inhibit the convection process and have a significant impact
on controlling the PR in winter (Zhao et al., 2006). Conversely, during the summer season,
meteorological instability prevails with low LTS values which result in high RH. This not only
causes enhanced AOD due to the water uptake and results in swelled hydrophilic aerosols
(Alam et al., 2010; Alam et al., 2011) but also affects the cloud and precipitation formation due
to the enhanced evaporation and convection. Additionally, Fig. 8-9 also shows evidently and
specifically during summer that the possible cause of positive AOD-CER correlation is the
negative AOD-CDNC correlation under unstable meteorology over all areas except Karachi.
As a result, Fig. 11 shows high (low) values of PR over all areas with a maximum over Gandhi
College (Karachi). The results show a high (low) approximation of PR over Gandhi College
(Karachi). Knowing that the rate of conversion of CDNC to precipitation is proportional to
CER (Wolf & Toon, 2014). Therefore, the high PR values are due to the growth of bigger cloud
droplets in summer. Further, apart from the reasons mentioned in the preceding sections, the
other justification for the differently perturbed aerosols, clouds, and precipitation pattern over
the study areas in summer is due to the entrance of southeast winds from the Bay of Bengal
passing across Gandhi College to Delhi and Lahore and the entrance of same winds from
Arabian sea to Pakistan through Karachi (Anwar et al., 2022).
Fig. 12 shows scatter plots of PR versus CDNC. The plot is colored with COT to examine the
impact of CDNC on PR for similar macrophysics. When CDNC is less, then the COTs are
sparse grow larger, form less reflective clouds, and precipitate faster (Kump & Pollard, 2008).
The same phenomenon seems true in our case. The results illustrate high PR (0.0007 mm/day)
values for clouds with COT ranging from 3 to 28 with  CDNC $< \sim 50$ cm$^{-3}$ and intermediate for
optically thick clouds and CDNC $> \sim 50$ cm$^{-3}$ in both seasons.

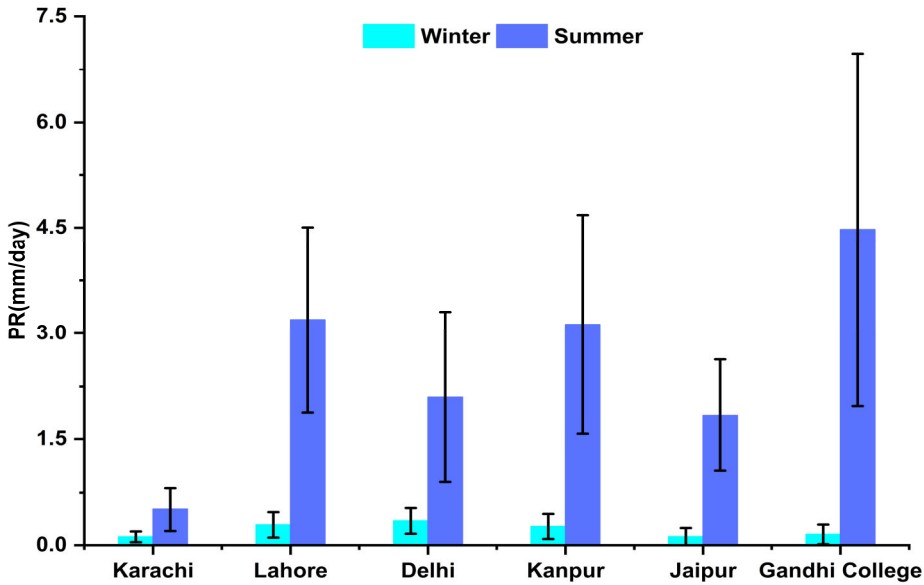


**Figure 11.** Mean Precipitation rate (PR) for the PCs in winter and summer season and SD
values with a 95% confidence interval.

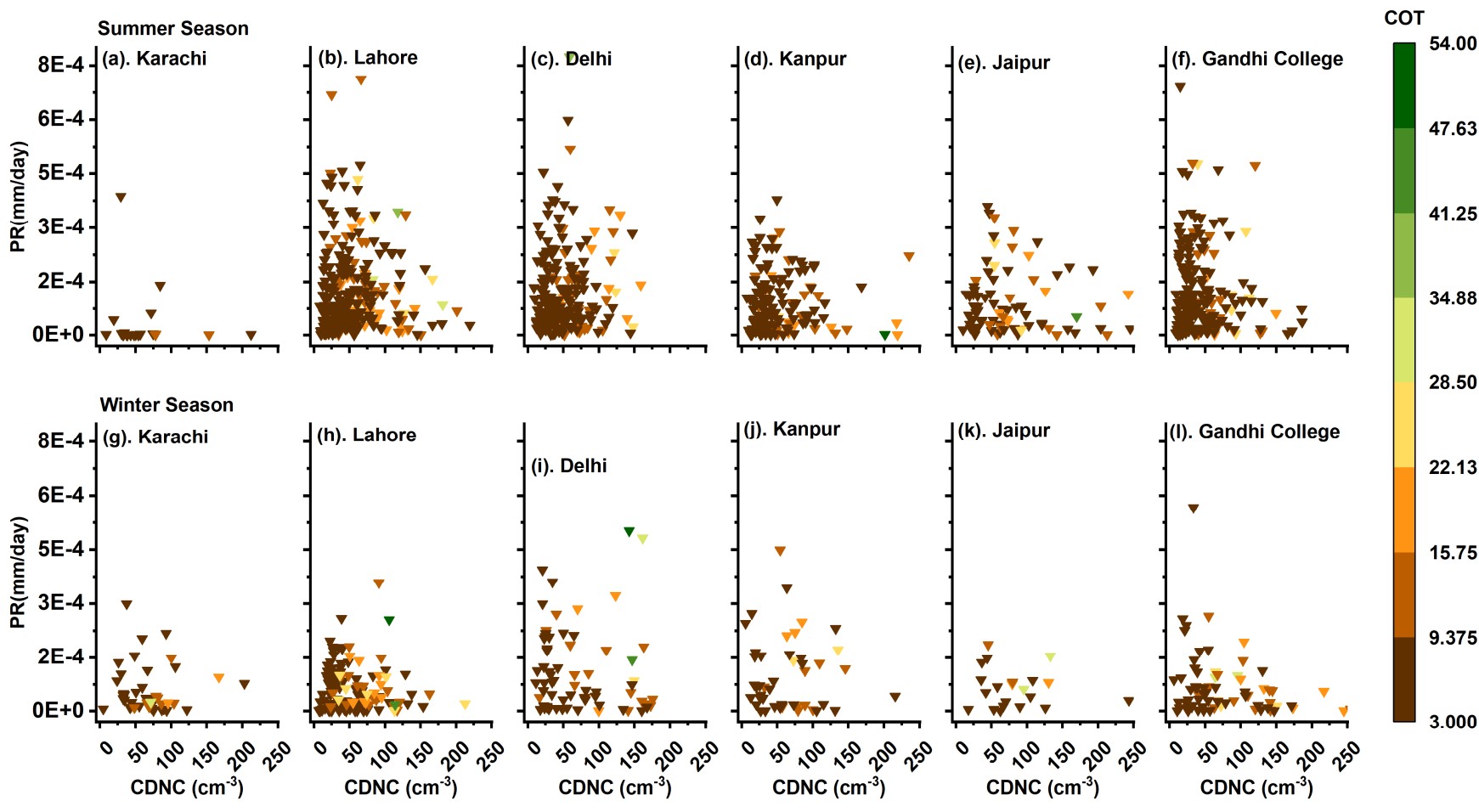


**Figure 12.** Scatter diagrams of PR (mm/day) versus CDNC (cm⁻³) in summer and winter seasons. color coding shows the COT of PCs.


Fig.13 shows the sensitivity ($S_o$) of PR to CDNC defined by $S_o = \left(-\dfrac{dln(PR)}{dln\,(CDNC)}\right)_{COT}$ for clouds
of low and intermediate thickness illustrated in Fig. 13 a and Fig. 13 b respectively. However,
sensitivity analysis for COT > 23 could not be performed due to the smaller number (0 to 04)
of available samples. In the sensitivity equation, the minus sign shows the suppression of
precipitation formation due to the increase in CDNC. Further, when $S_o$ is positive, the
correlation between PR and CDNC is negative; however, for negative $S_o$, PR and CDNC are
positively correlated. The results show peak values of $S_o$ i.e., $0.7 \pm 0.3$, $0.6 \pm 0.3$, $0.5 \pm 0.3$, and
$0.4 \pm 0.4$ over Jaipur, Delhi, Gandhi College, and Karachi respectively at intermediate values
of COT in winter, indicating the occurrence of lightly precipitating clouds. Referable to Fig.
13b, the low magnitude of $S_o$ $0.2 \pm 0.3$ and $0.08 \pm 0.2$ over Kanpur and Lahore respectively is
due to coagulation, in which precipitations are less sensitive to CDNC.

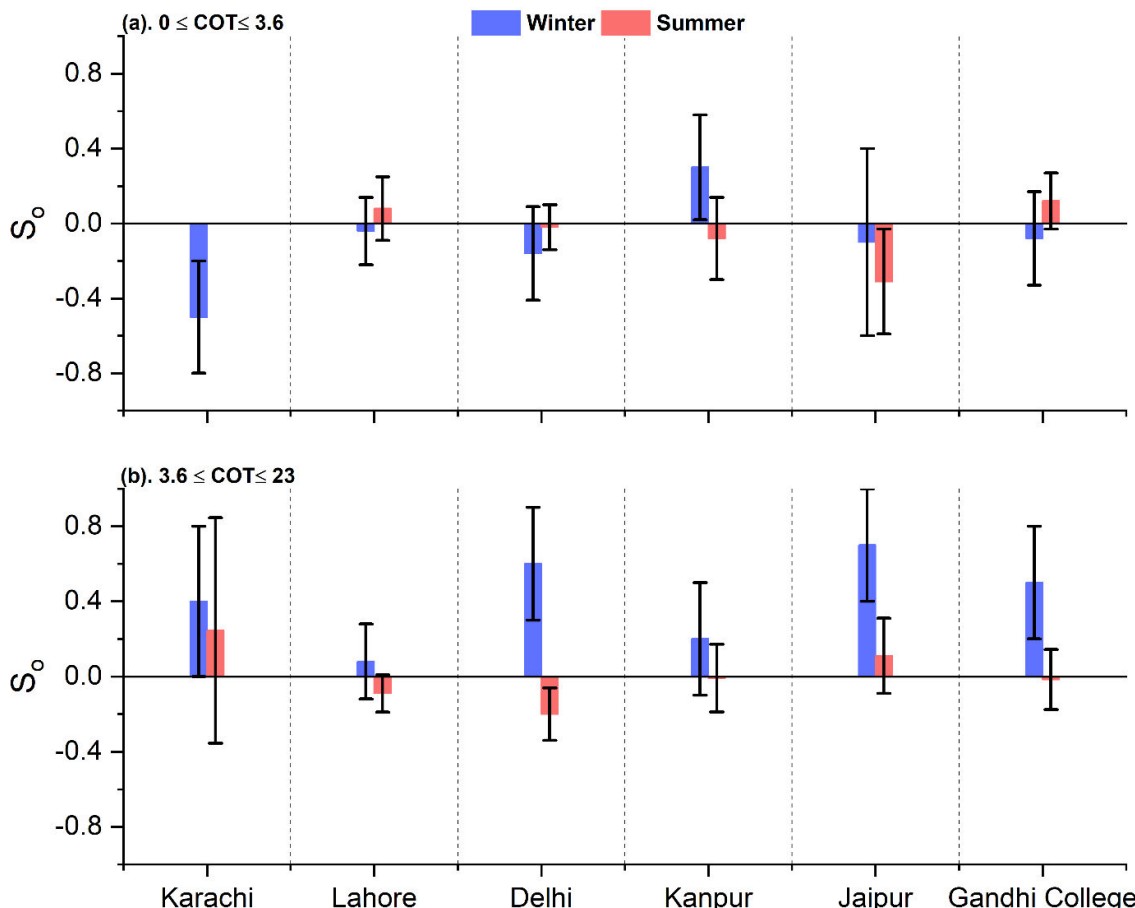


**Figure 13.** The sensitivity 'So' of precipitation rate (PR) for two bins of COT is shown in (a).
$0 \leq COT \leq 3.6$ and (b). $3.6 \leq COT \leq 23$.

## 4. Conclusion

In this study, the long-term (2001-2021) data retrievals from MODIS coupled with TRMM and NCEP/NCAR reanalysis-II datasets over the entire study area are compiled and analyzed for PCs and NPCs in the winter and summer season. The following are the main findings of this study.

A decadal decrease in AOD is observed over Karachi (-1.9%) and Jaipur (-0.5%). Meanwhile, AOD exhibits an increase over Lahore (5.2%), Delhi (9%), Kanpur (10.7%) and Gandhi College (22.7%). The LTS values are High (low) for NPCs (PCs) in winter and for PCs (NPCs) in the summer season. However, among all study areas, Karachi exhibits comparatively high LTS values in both seasons. Apart, the increase in RH% for PCs ranged from 33-57% in winter and from 25-45 % in summer. $\omega > 0$ for all NPCs in winter and $< 0$ for PCs in both winter and summer seasons.

In the winter season, a low frequency of cloudy days over Karachi and a high over Lahore and Gandhi College is estimated. Also, the high number of  PCs is estimated only over Lahore. In the summer season, out of the 74 % of the cloudy days, 60 % are PCs over Gandhi College. Similarly, most of the clouds over Lahore, Delhi, and Jaipur are PCs. Conversely, the lowest number of PCs (6 %) is found over Karachi.  The high-level PCs are identified in one bin of CTP (180< CTP < 440 hPa) over all study areas in winter. In the summer season, all three types of high-level and thick PCs have significant values of CF. The low-level PCs are identified as stratus clouds. Further, PDF values for CER $> \sim 15$ μm and CDNC $> \sim 50$ cm$^{-3}$ for NPCs and PCs are high (low) in summer (winter) over all areas except Karachi.

The AOD-CER correlation is good for PCs and weak for NPCs in the winter season. Also, the CER and CDNC values increase with the increase in LTS. The sensitivity value of FIE is high (low) for PCs (NPCs) in winter. Further, the magnitude of sensitivity of FIE (SIE) is low (high). Also, the sensitivity of TIE is a linear sum of the sensitivities of FIE and SIE. Further, ACI sensitivity values for PCs in summer are small, illustrating less dependency of CER on CDNC in thick clouds.

The high (low) PR values are observed in summer (winter). Further, high PR values for comparatively thin clouds with fewer CDNC $< \sim 50$ cm$^{-3}$ and intermediate for optically thick clouds and CDNC $> \sim 50$ cm$^{-3}$ are observed. Sensitivity values are small (high) for thick clouds in summer (winter).

Being one of the major source regions of anthropogenic aerosols across the globe, IGP offers interesting insights into the study of ACPI coupled with aerosol indirect effects. This study highlights that the aerosol-cloud relationship exhibits different behavior under different meteorological conditions, at coastal and inland locations. Thus, compared to other study areas, the stable atmospheric conditions due to the constant sea breeze weakened the ACI over Karachi, which resulted in a smaller number of CDNC, NPCs, and PCs. Further, our study also provides a very good platform for the detailed analysis of sensitivity tests of aerosol indirect effects and precipitation formation.

Although the sample size limits the study, the observed trends offer important insights that provide a foundation for future research. Therefore, further investigations with larger sample sizes are suggested to validate and extend these findings.

**Limitations and future recommendations:**

Although the current study is as thorough as possible, however, it has its limitations due to the topographical complexity of IGP, the lack of in-situ measuring instruments in Pakistan, and the intrinsic uncertainties associated with satellite-based data. Therefore, simulations of ground-based measurements along with satellite-based retrievals and calculation of cloud properties and CCN by different Community Atmosphere Model (CAM) and Weather Research and Forecasting (WRF) Models are recommended for deep insight into the various mechanisms of ACPI over IGP.

**Data Availability:** The MODIS and TRMM data can be obtained from the NASA Goddard Earth Sciences Data and Information Center (GES DISC) and can be retrieved from the websites: https://modis.gsfc.nasa.gov/data/ and https://gpm.nasa.gov/data . The reanalysis-II datasets are obtained from the website: https://psl.noaa.gov/data/gridded/data.ncep.reanalysis2.html . The processed datasets used in this work are available on reasonable request from the corresponding author.

**Author contribution:** NG processed and analyzed the data and wrote the original draft of the manuscript. KA proposed the Idea, supervised this work, and revised the manuscript. YL helped in revising the manuscript.

**Acknowledgment:** The authors gratefully acknowledge the NASA Goddard Earth Sciences Data and Information Services Center (GES DISC) for the provision of freely available data retrieved from MODIS and TRMM. We are also grateful to the NOAA Physical Sciences Laboratory (PSL) for free accessibility to (NCEP/NCAR) reanalysis-II datasets.

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
