# Peer review of "Influence of covariance of aerosol and meteorology on co-located precipitating and non-precipitating clouds over Indo-Gangetic Plains"

_EGUsphere, 2023_

## Author Comment (AC1)

**Reply to first reviewer on review of "Influence of covariance of aerosol and meteorology on co-located precipitating and non-precipitating clouds over Indo-Gangetic Plains" Gulistan et al.**

*We are very thankful to the anonymous referee for his/her expert opinion on our work which leads to the improvement of the manuscript. Below are the replies to the reviewer's comments, and indications of additions, modifications, or subtractions to the text under discussion.* ***We report the reviewer's comments in italic red, our responses in italic black, and the text added to the manuscript in roman Blue.***

1. *The paper asserts that the high loading of aerosols led to a high occurrence of precipitating clouds in summer. While high PR values in summer seem to have some associations with aerosols, the presentation seems to skirt around establishing a firm causality. Is it possible that some factors that influence both aerosols and precipitation simultaneously? Thus, this study may need to discuss the cause-and-effect relationship outlined in the study's conclusion.*

**Reply:** *We are thankful to the respected reviewer for constructive and insightful comments. In the revised manuscript we have discussed the meteorological parameters which may have significant implications on aerosols and precipitation formation. We also agree to discuss the cause-and-effect relationship outlined in the conclusion. For this reason, the following paragraph is inserted in the revision of the manuscript in the fourth line of section 3.4.3:*

The stable atmospheric condition with high LTS value in winter serves to inhibit the convection process and have a significant impact on controlling the PR in winter (Zhao et al., 2006). Conversely, during summer season, the meteorological instability prevails with low LTS values which result in high RH. This not only causes enhanced AOD due to the water uptake and resulted swelled hydrophilic aerosols (Alam et al., 2010; Alam et al., 2011) but also affects the cloud and precipitation formation due to the enhanced evaporation and convection. Additionally, Fig. 8-9 also show evidently and specifically during summer that the possible cause of positive AOD-CER correlation is the negative AOD-CDNC correlation under unstable meteorology over all areas except Karachi. As a result, Fig. 11 shows high (low) values of PR over all areas with maximum over Gandhi College (Karachi).

2. *The discussion may have a discussion of the uncertainties tied to the satellite datasets used in this study, as well as how the uncertainties may influence the AOD-CER correlation and the SIE.*

***Reply:*** *We are grateful to the reviewer for very nice comments on our manuscript. Yes, we agree that due to remote sensing nature, satellite retrievals have uncertainties which propagate to our results and findings and cause uncertainties in prediction of climate change. In this regard, to discuss the uncertainties in satellite retrievals and its propagation to AOD-CER and SIE, the following paragraph is inserted in the revised manuscript at the end of section 3.4.1:*

Recent advances in remote sensing led to cost-effective solutions and an increase in available data at various temporal and spatial resolution to bridge scientific gaps among different disciplines. While satellite-based retrievals have many advantages over in-situ and ground-based measurement such as broader regional coverage and enhanced spatial resolution, they are still prone to considerable uncertainties owing to the indirect nature of remote-sensing, retrieval algorithms, thermal radiance, infrequency of satellite overpasses, and cloud top reflectance (Hong et al., 2006; Tian et al., 2010; Hossain et al., 2006). In our study, apart from the aforementioned factors contributing to the uncertainty, any residual cloud contamination could also lead to biased retrieval of AOD. Likewise, satellite-based retrievals for cloud properties are crucial to understanding the pivotal role of clouds in climate and the role of clouds is still a dominant source of uncertainty in prediction of climate change. These, uncertainties in AOD and retrievals of cloud properties also propagate through the modeling process, potentially leading to less accurate climate predictions. Likewise, these uncertainties appeared to influence the findings in the current investigation. For instance, a limited correlation between AOD and CER is observed over Lahore, particularly in cloudier regimes as depicted in Fig. (5-6). This contrasts with robust impacted documented in the earlier studies (Michibata et al., 2014). However, high sensitivity of SIE is observed for PCs particularly in winter season indicating the delay in onset of precipitation and more retention of clouds.

3. *The study only considers a few locations, limiting the scope of the conclusions. The authors should discuss the implications of the relationships between clouds, aerosols, and precipitation over Indo-Gangetic Plains for other regions. For example, whether the high sensitivity value of the FIE in winter can be established to other regions.*

*Reply: Thank you for the detailed and useful comments. Following the suggestion, we have expanded the study to the eastern part by investigating Kolkata, Dhaka and Patna and covered the full Indo-Gangetic Plains. For complete investigation of ACPI over Kolkata, Dhaka and Patna, including the decadal and seasonal variations in aerosols, seasonal variations in meteorological parameters for PCs and NPCs, occurrence of*

*cloudy days and PCs, cloud types, influence of AOD on CER and CDNC at different atmospheric stability for PCs and NPCs in both summer and winters, indirect effects (FIE, SIE and TIE) and ACI for PCs and NPCs in summer and winter. Since, the number of figures increases significantly, we document the results for the eastern part as supplementary material. And as a result, in the revised manuscript the map of the study area is updated as follows.*

[Figure]

Fig. 1. Topography of the study area.

- *The following results for Kolkata, Dhaka and Patna are documented as supplementary material.*

Table 1S. Decadal percentage variations in average values of AOD over eastern part of IGP

|                          | Kolkata | Dhaka | Patna |
|--------------------------|---------|-------|-------|
| Total number of counts   | 1976    | 2018  | 2629  |
| Decadal change in AOD    | 18%     | 22.6% | 23.3% |

Table 1S shows the decadal change in AOD over Kolkata, Dhaka and Patna. Similar to Gandhi College, an increase is observed over all the three areas. Reason for the increase of aerosols include multiple sources of aerosols, human behavior, socio-economic

development at local and regional level, and unique topography for persistence and retaining of aerosols.

[Figure]

Fig. 1S. Probability density function (PDF) of AOD over study sites is shown (a) and (b) for summer and winter seasons respectively.

Fig. 1S shows the seasonal PDF values of AOD over all the three areas. The results indicate similar seasonal distribution functions. In both seasons PDF peaks for high values of AOD are observed over Patna showing high concentration of aerosols as compared to Kolkata and Dhaka.

[Figure]

Fig. 2S. Variations in lower tropospheric stability (LTS) over all study sites for PCs and NPCs in winter and summer seasons, the error bars show the standard deviation (SD) values.

Fig. 2S and Table 2S show the LTS conditions for PCs and NPCs. The high LTS values indicate more stable condition over Dhaka. Similarly, Table 2S shows the seasonal average values for other meteorological parameters. The results indicate high values of $T_{850}$, RH% and $\Omega$ over Patna in summer.

Table 2S. Meteorological parameters for PCs(NPCs) in summer and winter seasons. Maximum values are for both types of clouds shown in bold and minimum values are indicated as italic.

| | Winter Season | | | Summer Season | | |
|---|---|---|---|---|---|---|
| | $T_{850}$ (K) | RH% | $\Omega$ (m/s) | $T_{850}$ (K) | RH% | $\Omega$ (m/s) |
| Kolkata | **286.4±1.9 (286±1.86)** | 53.9±22.5 (42±14.5) | *0.004±0.1 (**0.08±0.09**)* | *295.7±1.6 (295.7±1.8)* | *70.3±12.5 (67.7±15.6)* | -0.15±0.07 (-0.14±0.07) |
| Dhaka | 286.2±1.5 (*285.2±1.9*) | *51.8±14.5 (50.9±13.8)* | *0.03±0.09(0.07±0.1)* | *294.6±1.1(294.6±1.2)* | *76.3±7 (75.9±8.4)* | *-0.13±0.07 (-0.1±0.08)* |
| Patna | *284.4±1 (284.3±2.4)* | **67.7±15.3 (56.6±13.9)** | **0.07±0.13**(*0.04±0.0.12*) | **297.5±1.1(297.5±1.4)** | **82.1±16.8(76.3±19.7)** | **-0.2±0.1 (*-0.17±0.1*)** |

[Figure]

Fig. 3S. Frequency of occurrence of total cloudy days (including PCs and NPCs) and only PCs is shown for both winter and summer seasons.

Fig. 3S shows the total number of cloudy days and the number of days on which PCs occurred. The high occurrence of clouds is observed over Kolkata and Dhaka in both summer and winter seasons. The high occurrence of PCs in summer is due likely to the significant impact of elevated aerosols with the southwesterly winds on the summer monsoons and occurrence of PCs. Therefore, Kolkata and Dhaka are of critical importance from perspective of aerosol loading and ACI (Dahal et al., 2022).

[Figure]

Fig. 5S. AOD-CER and AOD-CDNC correlation coefficient for PCs and NPCs over all study areas in winter season.

[Figure]

Fig. 6S. Same as Fig. 5S but in summer season.

[Figure]

Fig. 7S. The sensitivity metrices estimated for aerosol-cloud relationship using CDNC is shown in (a) FIE , (b) SIE, (c) TIE and (d) ACI.

Fig. 5S and 6S show the impact of AOD on CER and CDNC for PCs and NPCs in winter and summer respectively. The results indicate a positive and weak AOD-CER correlation for NPCs over all areas and for PCs over Kolkata and Patna in winters. Similarly, a positive and weak AOD-CDNC is observed over all areas for PCs. Likewise, Fig. 6S also illustrates weak correlation for both types of clouds in summer. As compared to other areas, the correlation analysis is less significant over Karachi, Kolkata, Dhaka and Patna. This can be attributed to the persistence of diverse aerosol types influenced by their coastal locations, different meteorology and the alternating inflow and outflow of easterly and westerly winds.

Fig. 7S(a) shows sensitivities for FIE in both seasons for PCs and NPCs. The results indicate high values of sensitivity FIE in winter season which is similar to the results for Karachi, Lahore, Delhi and Kanpur as shown in Fig. 10 a. This is attributed to high level of aerosol emission from residential heating and industrial activities. Furthermore, the results illustrate higher values of FIE in summer. This is attributed to the massive aerosol loading due to aerosol carried by winds and originated by anthropogenic activities and unstable meteorology.

*4. Although the manuscript mentions "the complications of aerosol-cloud-precipitation interactions over complex topography," it doesn't sufficiently explain how specific topographical features impact the ACI. A further discussion of potential topography impacts may strengthen the paper.*

**Reply:** *We are thankful to the reviewer for the detailed review of our manuscript. The unique topography of IGP influences the persistence of aerosols transported by the prevailing winds which influenced the ACI significantly. Therefore, following text is added at line 179 to discuss the impact of potential topography:*

IGP characteristically exhibits a diverse and massive pool of aerosols due to its unique topography. The western part of IGP is a coastal location and inlet for the westerly winds. Therefore, dry regions and Arabian sea in the west contribute dust, sea salt and water vapors to the region. The Himalayas in the north act as barriers to the winds, leading to the trapping of aerosols over the central part of IGP. Therefore, this region exhibits a high concentration of anthropogenic aerosols. Bay of Bengal in the east allows southeasterly winds to enter passing across Dhaka, Kolkata, Patna to Delhi and Lahore (Hassan et al., 2002; Anwar et al., 2022). The westerly and easterly winds traverse forested hilly terrain, rivers and lakes elevating humidity level and initiate the cloud formation by activation of the newly originated small aerosol particles as CCNs and cloud formation affecting the local microclimate.

**References**

Zhao, C., Tie, X., & Lin, Y.: A possible positive feedback of reduction of precipitation and increase in aerosols over eastern central China, Geo. Res. Let., 33, 11, https://doi.org/10.1029/2006GL025959, 2006.

Dahal, S., Rupakheti, D., Sharma, R. K., Bhattarai, B. K., & Adhikary, B.: Aerosols over the Foothills of the Eastern Himalayan Region during Post-monsoon and Winter Seasons, Aero. AQ. Res., 22, 4, 210152, https://doi.org/10.4209/aaqr.210152, 2022.

Hong, Y., Hsu, K. L., Moradkhani, H., & Sorooshian, S.: Uncertainty quantification of satellite precipitation estimation and Monte Carlo assessment of the error propagation into hydrologic response, Wat. Resc. Res., 42, 8, https://doi.org/10.1029/2005WR004398 , 2006.

Tian, Y., & Peters-Lidard, C.: A global map of uncertainties in satellite-based precipitation measurements, Geohys. Res. Let., 37, 24, https://doi.org/10.1029/2010GL046008 , 2010.

Hossain, F., Anagnostou, E. N., & Bagtzoglou, A.: On Latin Hypercube sampling for efficient uncertainty estimation of satellite rainfall observations in flood prediction, Comp. & geosc., 32, 6, 776-792, https://doi.org/10.1016/j.cageo.2005.10.006 , 2006.

Michibata, T., Kawamoto, K., & Takemura, T.: The effects of aerosols on water cloud microphysics and macrophysics based on satellite-retrieved data over East Asia and the North Pacific, *ACP*, *14*, 21, 11935-11948, https://doi.org/10.5194/acp-14-11935-2014, 2014.

Hassan, M. A., Mehmood, T., Liu, J., Luo, X., Li, X., Tanveer, M., & Abid, M.: A review of particulate pollution over Himalaya region: Characteristics and salient factors contributing ambient PM pollution. Atmos. Envi., 294, 119472, https://doi.org/10.1016/j.atmosenv.2022.119472 , 2002.

---

## Author Comment (AC2)

**Review "Influence of covariance of aerosol and meteorology on co-located precipitating and non-precipitating clouds over INdo-Gangetic Plains" Gulistan et al.**

*We are very thankful to the anonymous referee for his/her expert opinion on our work which leads to the improvement of the manuscript. Below are the replies to the reviewer's comments, and indications of additions, modifications, or subtractions to the text under discussion. We report the reviewer's comments in italic red, our responses in italic black, and the text added to the manuscript in roman green.*

**General overview,**

*The article studies the aerosol-cloud-precipitation interaction at six stations in the Indo-Gangetic Plains. The authors use the synergy of satellite observations and reanalyses to collocate information on cloud, aerosol, precipitation, and meteorological properties in winter and summer. Their analysis is based on the distinction between precipitating and non-precipitating clouds. Several plots show differences between different stations, seasons, and cloud types with different conclusions, e.g. that the lower tropospheric stability is higher for non-precipitating clouds in winter and lower for precipitating clouds. Another interesting result is that the precipitation rate maximum does not occur for the same cloud type when the cloud droplet number concentration is high or low. The study is comprehensive and it is appreciated that the authors analyzed their data set in different ways.*

- *Thank you very much for the positive comments and appreciation.*

*Nevertheless, the current study lacks a lot of information on the methodology, uncertainties and I am particularly concerned about a part of the study where the authors retrieve an indirect effect parameter with dependent datasets, therefore it is difficult to give credit to some result as it is now. I have detailed my various concerns below.*

*From page 14 on, there are no more line numbers, which makes it difficult to refer to the questions I want to ask. From page 14 on, I will refer to the page number, not the line number.*

*Major revisions:*

- *Methodology section: The satellite observations are associated with uncertainty, but it does not appear (except briefly mentioned on page 24). The results should be associated with the potential uncertainty. A detailed analysis of the propagation of uncertainty in the results should appear with a discussion of the implications.*

Reply: *Thank you for your thoughtful and kind suggestion. We comply with the reviewer's suggestion and include a discussion about the uncertainty in satellite retrievals and its propagation in our results. Further, in this regard the following passage is inserted (pls see also our response to referee 1) as last passage in section 3.4.1:*

Recent advances in remote sensing led to cost-effective solutions and an increase in available data at various temporal and spatial resolution to bridge scientific gaps among different disciplines. While satellite-based retrievals have many advantages over in-situ and ground-based measurement such as broader regional coverage and enhanced spatial resolution, they are still prone to considerable uncertainties owing to the indirect nature of remote-sensing, retrieval algorithms, thermal radiance, infrequency of satellite overpasses, and cloud top reflectance (Hong et al., 2006;Tian et al., 2010; Hossain et al., 2006). In our study, apart from the aforementioned factors contributing to the uncertainty, any residual cloud contamination could also lead to biased retrieval of AOD. Likewise, satellite-based retrievals for cloud properties are crucial to understanding the pivotal role of clouds in climate and the role of clouds is still a dominant source of uncertainty in prediction of climate change. These, uncertainties in AOD and retrievals of cloud properties also propagate through the modeling process, potentially leading to less accurate climate predictions. Likewise, these uncertainties appeared to influence the findings in the current investigation. For instance, a limited correlation between AOD and CER is observed over Lahore, particularly in cloudier regimes as depicted in Fig. (5-6). This contrasts with robust impacted documented in the earlier studies (Michibata et al., 2014). However, high sensitivity of SIE is observed for PCs particularly in winter season indicating the delay in onset of precipitation and more retention of clouds.

- *Related to uncertainty, there is no mention of the number of points used in the statistics. For example, Figure 6, there are a lot of regimes and I have some doubt that each regime has a large enough number of pixels to provide significant statistics. Figure 6 is an example,*

*but I have the same concern for all the other results. This is mentioned on page 29, but I would like to see the numbers.*

*Reply: Thank you for the good suggestions and your concern about the significance of our results. For the sake of large number of data points, we have analyzed daily averages for a period of 2 decades, data points less than 15 are not considered in further analysis.*

*Details of the data points for each regime are given below.*

**Table 1. Number of data points / observations for each CTP-COT joint histogram for both NPCs and PCs in both summer and winter over each area.**

**Winter (NPCs)**

| CTP (hPa) | Karachi Winter (NPCs) | | | Lahore Winter (NPCs) | | | Delhi Winter (NPCs) | | | Kanpur Winter (NPCs) | | | Jaipur Winter (NPCs) | | | Gandhi College Winter (NPCs) | | |
|---|---|---|---|---|---|---|---|---|---|---|---|---|---|---|---|---|---|---|
| 440 to <180 | 35 | 29 | 11 | 27 | 19 | | 19 | 17 | | 22 | 16 | | 29 | 19 | | 16 | 19 | |
| 680-440 | 60 | 63 | 19 | 79 | 34 | | 66 | 42 | 16 | 50 | 31 | 15 | 74 | 35 | 20 | 29 | 18 | |
| <800 to 680 | 570 | 107 | 18 | 293 | 183 | 21 | 357 | 258 | 22 | 438 | 228 | 20 | 376 | 173 | 48 | 380 | 92 | 17 |

**Winter (PCs)**

| CTP (hPa) | Karachi Winter (PCs) | | | Lahore Winter (PCs) | | | Delhi Winter (PCs) | | | Kanpur Winter (PCs) | | | Jaipur Winter (PCs) | | | Gandhi College Winter (PCs) | | |
|---|---|---|---|---|---|---|---|---|---|---|---|---|---|---|---|---|---|---|
| 440 to <180 | 17 | 15 | | 17 | 16 | 18 | 19 | 16 | 15 | 17 | 16 | | | 19 | | 20 | 17 | |
| 680-440 | 23 | 22 | 16 | 23 | 43 | 22 | 20 | 21 | 18 | 17 | 15 | 20 | 17 | 15 | 18 | 19 | 15 | 19 |
| <800 to 680 | 27 | 23 | | 60 | 53 | 27 | 26 | 34 | 22 | 27 | 16 | | 33 | 26 | 22 | 33 | 35 | 16 |

**Summer (NPCs)**

| CTP (hPa) | Karachi Summer (NPCs) | | | Lahore Summer (NPCs) | | | Delhi Summer (NPCs) | | | Kanpur Summer (NPCs) | | | Jaipur Summer (NPCs) | | | Gandhi College Summer (NPCs) | | |
|---|---|---|---|---|---|---|---|---|---|---|---|---|---|---|---|---|---|---|
| 440 to <180 | 52 | 76 | | 38 | 99 | 19 | 43 | 69 | 15 | 43 | 43 | 15 | 81 | 122 | 20 | 44 | 49 | 16 |
| 680-440 | 29 | 133 | | 55 | 136 | 21 | 54 | 80 | 18 | 52 | 51 | 18 | 38 | 90 | 15 | 24 | 40 | 20 |
| >800 to 680 | 162 | 400 | | 155 | 164 | | 110 | 66 | 41 | 91 | 64 | 27 | 27 | 82 | 17 | 44 | 34 | |

**Summer (PCs)**

| CTP (hPa) | Karachi Summer (PCs) | | | Lahore Summer (PCs) | | | Delhi Summer (PCs) | | | Kanpur Summer (PCs) | | | Jaipur Summer (PCs) | | | Gandhi College Summer (PCs) | | |
|---|---|---|---|---|---|---|---|---|---|---|---|---|---|---|---|---|---|---|
| 440 to <180 | 24 | 19 | | 26 | 63 | 16 | 24 | 68 | 21 | 18 | 50 | 16 | 16 | 25 | 15 | 35 | 76 | 20 |
| 680-440 | | 21 | | 31 | 88 | 20 | 39 | 71 | 34 | 17 | 54 | 22 | 19 | 24 | 20 | 23 | 47 | 17 |
| <800 to 680 | 31 | 33 | | 62 | 86 | | 42 | 45 | | 26 | 23 | | 15 | 22 | 16 | 29 | 40 | |

COT columns (for each city): 0-3.6 | 3.6-23 | 23 to >60

- *Methodology: There is no discussion of the collocation of the various products. How are CER and AOD collocated? MODIS does not retrieve AOD when a cloud is detected, did the authors look at the nearest pixel? If so, did they consider the potential 3D effect of clouds? How are MODIS and TRMM data collocated? Same questions with reanalysis (temporal and spatial resolution are not the same).*

*Reply: We are thankful to the reviewer his/her good suggestion and thoughtful comments. Following is the explanation/clarification regarding concerns of data collocation.*

- *Daily averages of AOD and CER are obtained from aerosol-cloud data product MOD08 of 1° X °l spatial resolution from MODIS TERRA with fraction of pixels that satisfy some conditions e.g., cloudy and clear (*MODIS Web (nasa.gov)*). Then statistical function is applied to align the data for both parameters on corresponding dates.*

- *Similar to the previous studies (Cheng et al 2017; Remer et al.,2005; Anwar et al., 2022), the AOD data are obtained from MODIS TERRA data product MOD08 using combined Dark target and deep blue algorithms. Further, following Anwar et al. (2022), data with AOD > 1.5 are excluded to avoid potential misidentification between aerosols and clouds.*

  *The potential 3D effect is not considered. However, the filtered data are tallied with MODIS-TERRA that use, true color images with corrected reflectance for both AOD > 1.5 and AOD < 1.5. For example, the true color image for AOD> 1.5 over Gandhi College, dated January 09, 2020 is given below in Fig (a).*

[Figure]

*And the true color image for AOD <1.5 on same location dated December 22, 2020 is given below Fig (b).*

[Figure]

*From the images it is clearly observed that for AOD> 1.5 (shown in Fig (a)) the cloud occurred over Gandhi college and for A0D< 1.5 (shown in Fig (b)) it is cloud free.*

- *Amin et al. (2009) validated and verified that the daily mean of PR from TRMM were coincident with the available ground-based records and confirmed its suitability for PR monitoring. The authors also concluded that MODIS-TERRA and TRMM data with their short retrieve time (daily) permit establishing a monitoring approach between both. Therefore, TRMM data retrievals are utilized to observe and analyze the PR.*

- *In this study the data retrievals from MODIS, TRMM and reanalysis are aligned through point-to-point collocation. In this type of collocation, the spatial coordinates*

*(latitude and longitude) are matched for the common points of datasets. And then align the temporal information of the observation at each point.*

- *Methodology: The authors consider temperature, LTS, RH, vertical velocity as meteorological parameters. It is not clear why they considered these parameters and not others.*

*Reply: We are grateful to reviewer 2 for his/her/their thorough reading of our manuscript. In this regard the following text is inserted on line 219 in the revised manuscript:*

Generally, LTS has relationships to factors such as temperature, humidity, wind patterns, and atmospheric pressure over extended periods. It is also widely acknowledged that atmospheric stability, temperature, RH and wind speed and direction play a significant role in cloud formation (Yang et al., 2015; Tao et al., 2012). Therefore, the influence of long-term variations in the said meteorological parameters are considered in the current study.

*I suspect that the authors considered only liquid clouds for their analysis, but this is not mentioned in the manuscript. Did the authors filter their data set, and if so, how?*

*Reply: Many thanks to the reviewer for constructive and insightful comments. Yes, in the current study only liquid clouds are considered and it is mentioned on line 93 that the analysis is done for low level clouds which means liquid clouds over IGP. Fortunately, aerosol-cloud data product, MOD03 version 6.1 of MODIS-TERRA allows the retrieval of data for liquid and ice clouds as separate variable in the search tabs, available on [https://giovanni.gsfc.nasa.gov/giovanni/](https://giovanni.gsfc.nasa.gov/giovanni/) .*

- *The authors based their study on 6 stations, but they only use satellite and reanalysis datasets, so I wonder why they only focus on 6 stations and not do a full analysis of the region. The eastern part of the region is not considered, and this is unfortunate if satellite observations are considered. I expected a comparison with ground observations to explain why the sites were chosen, but there is none. Therefore, I would suggest explaining why the full map of the region is not considered.*

*Reply: Thanks for the excellent comment. The study is extended to the eastern part by including Kolkata, Dhaka and Patna as study sites. For detailed analysis please see our response to a similar comment of referee 1.*

- *FIE, SIE, and TIE parameters are based on dependent records. CDNC is derived from CER and CLWP, and CLWP is derived from CER and COT. Therefore, it is not possible to infer the different parameters. To study these effects, we should consider only independent datasets. Therefore, Figures 8, 9, 10 and the related discussions and conclusions, I have doubts about them.*

*Reply: Thank you for the nice comment. Of particular importance is the fact that aerosols may serve as cloud condensation nuclei (CCN). Increased aerosol concentrations may thus increase cloud droplet number concentration (CDNC), enhancing the cloud albedo (Twomey, 1974), and enhancing cloud lifetime and liquid water content by lowering the collision/coalescence rate (Albrecht, 1989). These so-called "indirect effects" of aerosols on liquid water clouds are referred to as the cloud albedo or first indirect effect and the cloud lifetime or second indirect effect. Therefore, according to the valuable comment of the respected reviewer, CDNC is dependent parameters. However, it has a pivotal role in the indirect effects.*

- *The article lacks quantification in most of the paragraphs, which makes the analysis difficult to follow because I sometimes do not know what is being referred to.*

*Reply: Thank you for the kind suggestion. We have taken the suggestion and added quantitative discussion in the revision. For example, the following lines are revised in section 3.3.1:*

- Most of the identified PCs are formed in the two bins of CTP ($180 < CTP < 440$ hPa) and ($440 < CTP < 680$ hPa) with CF values ranging from 0.8 to 1.0. The results suggest low values of CF for the low-lying thick NPCs over all study areas.
- Similarly, the type of PCs in both summer and winter season that occurred with CF ~1.0 include cirrus and cirr-stratus.
- In addition, among all the estimated low-level PCs, cumulus and strato-cumulus exhibit good CF values (0.7) over Kanpur and Gandhi College.

  *The following text is revised in section 3.4.3:*

- The results illustrate high PR (0.0007 mm/day) values for clouds with COT ranging from 3 to 28 with CDNC < ~ 50 cm$^{-3}$ and intermediate for optically thick clouds and CDNC > ~ 50 cm$^{-3}$ in both seasons.
- However, sensitivity analysis for COT > 23 could not be performed due to less number (0 to 04) of available samples. In the sensitivity equation the minus sign shows the suppression of precipitation formation due to the increase in CDNC. Further, when $S_o$ is positive, correlation between PR and CDNC is negative; however, for negative $S_o$, PR and CDNC are positively correlated. The results show peak values of $S_o$ i.e., $0.7 \pm 0.3$, $0.6 \pm 0.3$, $0.5 \pm 0.3$, and $0.4 \pm 0.4$ over Jaipur, Delhi, Gandhi College and Karachi respectively at intermediate values of COT in winter, indicating the occurrence of lightly precipitating clouds. Referable to Fig. 13b, the low magnitude of $S_o$ $0.2 \pm 0.3$ and $0.08 \pm 0.2$ over Kanpur and Lahore respectively is due to coagulation, in which precipitations are less sensitive to CDNC.

**Minor revisions:**

- *l.11, "different physical mechanisms", can the authors specify the different physical mechanisms?*

*Reply: Thank you for the comment. Yes, the different physical mechanisms are mentioned in revision of manuscript on line 93 as follows:* (condensation/droplet growth and precipitation rate).

- *l.11 "systematically analyze", what does systematically mean?*

*Reply: Here 'systematically analyze' means an organized approach of investigation or set of procedure to gather, organize , analyze and interpret information.*

- *keywords, "Aerosol option depth" -> "aerosol optical depth"*

*Reply: Corrected. Thank you.*

- *Figure from the abstract, I am not sure if the figure helps to understand the abstract, it is rather a lot of information with parameters not yet defined (LTS, CER, AOD), I suggest removing it.*

*Reply: We comply with the good suggestion of respected reviewer and removed the figure.*

- *Introduction: citations are missing, I suggest adding citations. For example, the first two sentences should have citations.*

*Reply: Thank you for suggesting the addition of citations. In the revised manuscript the following citation is inserted for the first two lines:* (Romero et al., 2021)

- *l. 66, aerosols can also act as ice nucleating particles but this is never mentioned in the article. Did the authors consider this? I think it should at least be mentioned in the introduction and emphasize that only liquid clouds are relevant for the analysis.*

*Reply: The good suggestion of the reviewer is implemented.*

- *l. 67, "The decrease in CDNC and increase in CER increases the probability of precipitation rate (PR)". Stevens and Feingold (2009) have shown that you can actually have the opposite effect: An initial inhibition of precipitation from aerosols can lead to increased precipitation later. The region is affected by large precipitation and this may be an effect that the authors did not consider. I suggest adding a discussion about it.*

*Stevens, B., & Feingold, G. (2009). Untangling aerosol effects on clouds and precipitation in a buffered system, Nature, 461(7264), 607-613.*

*Reply: We are thankful to the reviewer for pointing out a very useful research work by Stevens, B., & Feingold, G. (2009). Relevant to our study, the following text is added on line no. 67:*

Conversely, Stevens and Feingold (2009) have shown that initially, more sea salt carried by high wind speed inhibit the precipitation formation. However, the same sea spray tends to seed the coalescence by producing larger CER that led to enhanced precipitation.

- *l. 82: "Twomey effect", I think it would be best to describe the effect before mentioning it.*

*Reply: The good suggestion is implemented by adding the following text on line 82:*

decrease (increase) in CER with aerosol loading Twomey effect (anti-Twomey effect) over the monsoon  (weak and moderately intensive monsoon) regions.

- *l. 83: "anti-Twomey effect", I do not know this effect, can the authors describe it?*

*Reply: The increased aerosols can reverse the Twomey effect in water clouds. In the anti-Twomey effect, with a potential decrease in CDNC, droplets of larger size are formed with the increased aerosol loading leading to the decreased cloud albedo (Khatri et al., 2022).*

- *l.87: "FIE", the acronym is not defined before.*

*Reply: Your good suggestion is implemented.*

- *l. 87, the Twomey effect refers to the change in cloud radiative properties and not to the cloud droplet size. Also McCoy et al., (2018) may not be the best and the citation from Twomey may be more relevant*

  *Twomey, S. (1977). The influence of pollution on the shortwave albedo of clouds. Journal of the atmospheric sciences, 34(7), 1149-1152.*

*Reply: In the Twomey effect a large number of smaller cloud droplets are formed. Smaller droplets scatter sunlight more effectively than larger droplets which can result in a cloud that appears brighter and reflects more solar radiation back into space. Therefore, the authors agree with the reviewer that Twomey effect refers to the change in cloud radiative properties. Further, the reference is updated to insert the suggested reference.*

- *l. 111, I suggest including the names of the cities on the map.*

*Reply: Thank you. Good suggestion of the reviewer is implemented as follows.*

[Figure]

Fig. 1. Topography of the study area.

- *Figure 1 caption: "Geographical map" -> "topography".*

Reply: *Thank you. The good suggestion is implemented as shown figure caption in reply to previous comment.*

- *Figure 1: What is the data for topography? Some regions are covered and some are not. I suggest removing Figure 1 and adding the points on Figure 2 (with the names of the stations) since there is no mention of topography in the article.*

Reply: *In response to the good suggestion of the reviewer instead of removing Fig.1, the following explanation is added about the topography at line 179:*

IGP characteristically exhibits a diverse and massive pool of aerosols due to its unique topography. The western part of IGP is the coastal location and inlet for the westerly winds. Therefore, dry regions and Arabian sea in the west contribute dust, sea salt and water vapors to the region. The Himalayas in the north act as barriers to the winds, leading to the trapping of aerosols over the central part of IGP. Therefore, this region exhibits a high concentration of anthropogenic aerosols. The Bay of Bengal in the east

allows southeasterly winds to enter passing across Dhaka, Kolkata, Patna to Delhi and Lahore (Hassan et al., 2002; Anwar et al., 2022). The westerly and easterly winds traverse forested hilly terrain, rivers and lakes elevating humidity level and initiate the cloud formation by activation of the newly originated small aerosol particles as CCNs and cloud formation affecting the local microclimate.

- *l. 122, "resolution of x to study atmospheric…", change x to the correct value.*

Reply: Thank you for your correction. Changed to ($1°$) at line 122.

- *l.125, CDNC and CLWP are not direct products of MODIS. They are defined later but should not appear here.*

Reply: Thank you. your kind suggestion is implemented.

- *l. 126 "Data with AOD>1.5", with the threshold the authors avoid misidentification of clouds as aerosols and not the reverse as stated in the article. Is there a threshold to avoid misidentification of clouds as aerosols?*

Reply: Thank you for the nice comment. Although a detailed reply is given to such comment in the major revisions. However, it is further explained as follows.

The following table shows the threshold values AOD for classification aerosols into different types (B AL-Taie et al., 2020).

| Aerosol type | Aerosol optical depth (AOD) | Angstrom exponent(AE) |
|---|---|---|
| Maritime | < 0.3 | 0.5-1.7 |
| Dust | > 0.4 | < 1.0 |
| Urban | 0.2-0.4 | > 1.0 |
| Desert dust | > 0.45 | 0.4-2.0 |
| Biomass burning | > 0.7 | > 1.0 |

Generally, the value of AOD ranges from 0.05 to 1 over the remote ocean and 2.0 to 5.0 during the time of heavy pollution smoke and dust (Petal et al., 2016).

Therefore, idea of excluding AOD > 1.5 to avoid clouds as aerosols is taken from a recent study (Anwar et al., 2022), which may not be the threshold value for this purpose.

- *Equation 1, square root does not go all the way.*

*Reply: Thank you. Equation 1 is corrected as follows:*

$$\text{CDNC} = \left(\frac{B}{\text{CER}}\right)^3 * \sqrt{(2 * \text{CLWP} * \gamma_{\text{eff}})}$$

- *LTS equation (line 144), is not numbered.*

*Reply: Thank you. LTS equation is numbered as (3)*

- *l. 144, \theta_{0} -> \theta_{1000}*

*Reply: Thank you. The correction is done as follows.*

$$\text{LTS} = \theta_{700} - \theta_{1000}$$

- *l. 150, PR is defined for precipitation rate but is not an instrument and for Precipitation Radar it is not mentioned.*

*Reply: Thank you. Corrected.*

- *l. 150, TMI is not defined.*

*Reply: Thank you. TMI is defined as follows:*

TRMM Microwave Imager (TMI).

- *Methodology section: is both Aqua and Terra for MODIS used?*

*Reply: Thank you for the detailed and thorough reading of the manuscript. In the methodology section it is already mentioned that "level 3 aerosol-cloud data product MOD08" which means MODIS-TERRA. While the same data product of MODIS AQUA is named MYD08.*

- *l. 187, "is similar", the authors state the opposite afterwards so I would remove the "similar"*

*Reply: Thank you. The good suggestion is implemented.*

- *lines 195 and 196, citations are missing.*

*Reply: Thank you. The following citation is added.*

(Sun & Ariya, 2006).

- *l. 199, Why does Karachi have lower values?*

*Reply: Thank you for the insightful comment. Reason for lower AOD values is inserted on line 199 as follows.:*

Ali et al., 2020 associated the low AOD values over Karachi to the westerly and southwesterly winds currents at tropospheric level. However, the decreasing trend in AOD over the coastal city may also be attributed to the variations in other meteorological parameters like T and RH.

- *l. 201, "illustrate pristine atmosphere", I suggest adding "comparatively".*

*Reply: Thank you. Nice suggestion of the reviewer is implemented.*

- *lines 200-206, it would be better to quantify with the median to compare different regimes.*

*Reply: Thank you for your advice. This has been very useful. Per your good suggestion lines 200-206 are revised as follows.*

As compared to summer season, the pattern of PDF in winter is significantly different as shown in Fig. 3b. The low value of PDF (0.5) for the high value of AOD (0.9) over Karachi illustrates a comparatively pristine atmosphere. Similarly, the PDF peaks for Lahore, Delhi and Jaipur (0.7, 0.7 and 0.8) indicate comparatively high AOD over Delhi. Likewise, the distribution over Kanpur and Gandhi College is similar illustrating similar values of AOD (1.1and 1.2 respectively).

- *l. 204, before the new sentence, the authors compare with the other PDF, I think it should be a new paragraph with the description.*

*Reply: Thank you for useful suggestion. The correction is made per your insightful suggestion. on line 204, the new sentence is revised as new paragraph as follows:*

Few authors attributed the reduced values of AOD in winter season to the wet scavenging and suppressed emission of aerosols from earth surface (Alam et al., 2010; Zeb et al., 2019). However, in our case, the low (high) values in winter (summer) are associated to dispersion of fine (course) mode particles due to the variations in meteorological conditions.

- *l. 204, "winter season is the wet scavenging", it contradicts with Fig 5 and Fig. 11 for which summer as more precipitation. Can the author explain?*

*Reply: Thank you for your good remarks. The reason is explained as follows:*

However, in our case, the low (high) values in winter (summer) are associated to dispersion of fine (course) mode particles due to the variations in meteorological conditions.

- *From Fig 3, AOD in winter is not smaller for Jaipur and Kanpur (it does not look like it). Any reason for this difference?*

*Reply: We comply with the useful suggestion of the reviewer and explained the same with quantification of AOD values and reason for high values AOD over Kanpur and Jiapur as follows:*

Similarly, the PDF peaks for Lahore, Delhi and Jaipur (0.7, 0.7 and 0.8) indicate comparatively high AOD over Delhi. Likewise, the distribution over Kanpur and Gandhi College is similar illustrating similar values of AOD (1.1and 1.2 respectively). These high values of AOD are attributed to the high emission of anthropogenic aerosols at local and regional level over the central part of IGP (Delhi, Jaipur, Kanpur and Gandhi College).

- *Table 1 "Total number of counts", "counts", is it pixels?*

*Reply: This is not pixels but the number of observation/ days.*

- *Figure 3, it is difficult to distinguish the different points and colors especially between Karachi and Gandhi College,*

*Reply: Thank you for your good suggestion. Fig.3 is revised as follows:*

[Figure]

- *l. 219, "estimation of", I think should be removed.*

Reply: Thank you so much. The correction is made per good suggestion of the reviewer.

- *l. 221-222, "the potential for vertical convection… precipitation formation", I do not understand this part, can the author rephrase?*

Reply: Thank you for the very useful remark. The good suggestion is implemented by rephrasing line 221-222 as follows:

LTS to determine the lower atmospheric stability and instability that influence the process of cloud and precipitation formation through its significant implications on evaporation and convection of the air parcel,

- *l. 235, "is relatively high", is it compared to PCs or to summer?*

Reply. Thank you for your time and thorough reading of our manuscript. This is compared to NPCs in summer season.

- *l.236 "The increase", which increase? I am not sure what it refers to. (same for line 242)*

Reply: Thank you for the useful and detailed comments. Per your comment line 236 is revised as follows:

The increase in RH% for PCs during winter ranged from (60±5)% to (72±5)%.

Similarly, line 242 is revised as follows:

Also, the increase in RH% during summer ranged between 25-45 %.

- *l. 236 "33-57%", I do not find these values in the Table.*

*Reply: Thank you for your comment. Line 236 is already corrected and revised. Please refer to previous comment.*

- *Sometimes RH is referred to as RH% and sometimes RH.*

*Reply: Thank you so much for your useful remarks and suggestions. The correctio is made per your good suggestion.*

- *l. 242, "suitable thermodynamical conditions" can the authors say more about this and add a citation?*

*Reply: To comply with the useful suggestion of the respected reviewer the following passage is inserted on line 242:*

The reason for the high values of wv and RH% is mainly the suitable thermodynamical conditions such as evaporation and convection due to the high temperature of earth surface and air (Sherwood et al., 2010). The results show high values of RH% $72 \pm 5$ $(71.6 \pm 3)$ in winter (summer) season for PCs over Gandhi College. Conversely, notable fluctuations in RH% are observed over the coastal city, Karachi, with values of $70 \pm 13.9$ $(68.4 \pm 6.7)$ in winter (summer).

- *l 244, Gandhi's college has a higher value of RH. I was expecting Karachi because it is closer to the coast. Can the author add a discussion about this?*

*Reply: Thank you. Line 244 is revised per useful remark of the reviewer as follows:*

The results show high values of RH% $72 \pm 5$ $(71.6 \pm 3)$ in winter (summer) season for PCs over Gandhi College. However, high values of standard deviation show notable fluctuations in RH% over the coastal city, Karachi, with values of $70 \pm 13.9$ $(68.4 \pm 6.7)$ in winter (summer).

- *Table 2, did the author consider the mean? I would suggest considering the median since we do not expect Gaussian distributions.*

*Reply: Thank you for the kind suggestion. We considered (mean ± SD) so that the fluctuated values can also be examined.*

- *page 14 "the frequency of occurrence of precipitable clouds" is it the frequency of occurrence relative to the total or to cloudy pixels?*

*Reply: This is the frequency of occurrence relative to total clouds (both precipitable and nonprecipitable).*

- *page 14, the authors apply filters to avoid overestimation (COT and CF> 5), but I wonder if this does not lead to underestimation.*

*Reply: Thank you giving useful suggestions to improve our manuscript. The correction is made per your good suggestion. This is not (COT and CF> 5) but (COT ~ > 5) for PCs only. Thank you.*

- *page 14, some discussion of the results is missing.*

*Reply: Thank you for your good suggestion. To comply with your useful suggestion the following lines are added in the discussion on page 14:*

Chen et al. (2018) suggested the COT to be the effective measure for assessing the clouds and potential for precipitation. In our case, to avoid any overestimation, the COT data are aligned with PR data on corresponding dates and then filtered to include COT ~ > 5 for PCs.

- *Fig. 6, I think the authors are not showing a joint histogram as stated in the article but rather the CF for different regimes of COT and CTP, can it be explained in more detail what is shown here? Is CF averaged?*

*Reply: Mean values of CF were calculated for all regimes. However, in response to one of the following comments, median of CF is calculated and therefore, fig. 6 is revised in the revised manuscript.*

- *Page 16: Is CF>0.7 a threshold used for the entire paragraph to state that it is "high CF"? if so, it should not be in parentheses but rather explicitly explained.*

*Reply: Thank you for the useful suggestion. No CF> 0.7 is not the threshold value. Further, the quantification of CF and related discussion is revised as Fig. 6 is revised in response to one of the following comments for median values of CF. The suggestion is also implemented, and CF value is explicitly mentioned as follows:*

The results exhibit noticeable differences in the pattern of cloud regimes over all study areas. The diverse CF values are observed in winter and summer seasons for NPCs and PCs over Karachi. In winter season, only stratus NPCs ($23 < COT < 60$, $800 > CTP > 680$ hPa) are dominant with CF ~ 0.9. While, in summers, high value of CF ~ 0.9 for low and intermediate thickness of high-level clouds such as Cirr-Stratus NPCs ($3.6 < COT < 23$, $180 < CTP < 440$ hPa) are observed. Similarly, the type of PCs in both summer and winter season that occurred with CF ~1.0 include cirrus and cirr-stratus. The relatively reduced value of CF for thick NPCs in winters and PCs in summers is attributed to the low values of AOD and high values of LTS. The results depicted slight differences and similarities in CF values for thick and thin NPCs respectively in winter season for all areas except Karachi. Besides, the high-level PCs are identified in the two bins of CTP ($180 < CTP < 440$ hPa) and ($440 < CTP < 680$ hPa) over all study areas. The formation of these similar types of PCs in winters are associated with the similarities in $\Omega$, LTS values and aerosols concentration.

- *Page 16, ($23 < COT < 60$, $CTP > 680$), should read ($23 < COT < 60$, $800 > CTP > 680$).*

*Reply: Thank you for your useful feedback on our manuscript which improved our manuscript. The correction is made on page 16 as follows:*

($23 < COT < 60$, $800 > CTP > 680$).

- Page 16, "Similarly, in winter season the type of PCs…" why cirrus & cirro-stratus not included with CF>=0.9.

*Reply: Thank you for useful advice. The discussion is revised in response to one of the following comments after revision of figure, and cirrus and cirro-stratus are included with median value of CF ~1.0 . Please refer to one of the following comments explained with figure.*

- *Page 16, "less significant values", do the authors mean "lower"?*

*Reply: Thank you the valuable comment. Yes, here, "less significant values", means low values of CF.*

- *Page 16: The paragraph starting with "Also, in summer…" it is difficult to follow this paragraph, I suggest changing the presentation of the paragraph.*

*Reply: Thank you for the valuable feedback on our manuscript. The said paragraph is revised per your good suggestion as follows:*

Likewise, in summer season, the matrices of PCs and NPCs exhibit a wide range of cloud types. However, the CF values are comparatively high for PCs. Most of the identified PCs are formed in the two bins of CTP (180< CTP < 440 hPa) and (440< CTP < 680 hPa) with CF values ranging from 0.8 to 1.0. The results suggest low values of CF for the low-lying thick NPCs over all study areas. Moreover, the results illustrate a more frequent occurrence of all the three types of thick NPCs in one bin of COT (23 < COT< 60) and all the three types of high-level NPCs for CTP (180 < CTP < 440 hPa) over Delhi, Kanpur, and Gandhi College. Therefore, these are considered the cloudiest regimes. Besides, contrasting regional variations are also observed in PCs. The maximum CF values for all types of PCs are observed over Kanpur and Gandhi College. Similarly, relatively good values of CF in a bin of COT (23 < COT< 60) and a bin of CTP (180 < CTP < 440 hPa) over Lahore, Delhi, and Jaipur depict the frequent occurrence of thick and high-level PCs respectively. In addition, among all the estimated low-level PCs, cumulus and strato-cumulus exhibit good CF values (0.7) over Kanpur and Gandhi College. The formation of thick clouds can be attributed to the enhanced convection process due to the atmospheric instability.

- *Table 3, ">800 to 680" should it be "<800 to 680"?*

*Reply: The correction is made in Table 3 per good suggestion as follows:*

<800 to 680

- *Figure 6, Is the mean CF shown? If so, the number of points and the SD should also be shown. I would also suggest showing the median instead of the mean.*

*Reply: Thank you for the valuable suggestion. Median instead of mean is calculated and Fig 6 is revised as follows:*

- *Page 19 "depicted an approximately similar values", did the authors perform a statistical test to infer this conclusion?*

*Reply: We are thankful and appreciate the thorough reading of our manuscript. Yes, the statistical analysis is done by applying the 'probability distribution function' (PDF). Then the conclusion is made on the basis of quantification of PDF obtained.*

- *Page 19 "The low number of CDNC", there is no information on CDNC in Figure 6*

*Reply: Thank you for your valuable time and suggestions. The correction is made on page 19 for the said sentence. Actually, this sentence is for the results shown in figure 7 not 6. Thank you again.*

- *page 21 "detailed linear regressed slopes", what is meant by "detailed".*

*Reply: Here 'detailed' means the slope with the trend line equation, along with values of regression and correlation coefficient.*

- *page 21 "correlation is good for PCs and weak for NPC", what is the criteria and threshold to determine if it is good or not? In both cases, r2 looks low.*

*Reply: Thank you for detailed feedback on our manuscript. It is observed in winters that the value of correlation coefficient 'R' is higher for PCs than that for NPCs. Further, it is good correlation if R >~ 0.5, either positive or negative.*

- *page 21 the positive AOD-CER correlation, what exactly does it mean physically ? Why should droplets be larger in the presence of aerosols?*

*Reply: Thank you for you useful and valuable remarks. The AOD-CER correlation can either be positive or negative. Positive AOD-CER means increasing CER with increasing aerosol loading. Few authors associated the positive (negative) AOD-CER correlation to the unstable (stable) meteorology and moist (dry) regions (Yuan et al., 2008).*

- *Figure 10 "the error bars show the standard deviation" the plot represents sensitivities, so I am not sure to understand what standard deviation is being retrieved here.*

*Reply: Thank you for the nice comments. Due to the variations in local and regional meteorology, the mean value of sensitivity fluctuates between maximum and minimum. For this reason, the standard deviation is retrieved here.*

- *Page 27, I am not sure what the authors mean by "approximation", is it uncertainty ?*

*Reply: Since satellite-based retrievals are prone to considerable uncertainties owing to the indirect nature of remote sensing. These uncertainties in retrievals also propagate through the modeling process, potentially leading to less accurate climate predictions. Therefore, the word "approximation", is used in the discussion .*

- *Page 27 "Fig. 12 shows scatter plots of…" the paragraph lacks quantifications,*

*Reply: Thank you. Quantification is added per your useful feedback and suggestion as follows:*

Fig. 12 shows scatter plots of PR verses CDNC. The plot is colored with COT to examine the impact of CDNC on PR for similar macrophysics. When CDNC are few, then the COT are sparse that grow larger, form less reflective clouds and precipitate faster (Kump & Pollard, 2008). The same phenomenon seems true in our case. The results illustrate high PR (0.0007 mm/day) values for clouds with COT ranging from 3 to 28 with CDNC $< \sim 50$ cm$^{-3}$ and intermediate for optically thick clouds and CDNC $> \sim 50$ cm$^{-3}$ in both seasons.

- *Fig. 11 caption, is the mean shown ?*

*Reply: Thank you. Yes, the bars show the mean values of PR. Caption of Fig. 11 is revised in the revised Manuscript as follows.*

Fig. 1. Mean Precipitation rate (PR) for the PCs in winter and summer season and SD values with 95% confidence interval

- *l. 30 "Also the frequently occurred…" it should be rephrased.*

*Reply: sorry "Also the frequently occurred ….." could not be found on line 30 or page 30.*

- *page 31, the ladsweb website does not work,*

*Reply: Thank you for useful comment. The URL is corrected as follows:*

*https://giovanni.gsfc.nasa.gov.giovanni/*

- *Bibliography: multiple references are not correctly written in the bibliography (some doi are underlined, some are not),*

*Reply: Thank you for the valuable remarks and suggestion. For implementation of your suggestion refer to the next comment please.*

- *Bibliography: some names are not outputed correctly, for example: "Thomas, A., Kanawade, V. P., Sarangi, C., & Srivastava, A. K. J. S. o. t. T. E"*

*Reply: Thank you for detailed and useful feedback which helped to improve our revised manuscript. Your good suggestions about bibliography are implemented in references as follows:*

[revised manuscript text omitted]

- *I suggest removing words that are unnecessary (meaningful (179), completely (l. 225), evidently (l. 228)... )*

*Reply: Thank you. Your useful and valuable suggestions are implemented and the unnecessary words such as meaningful (line 179), completely (line. 225), evidently (line 228,…) are removed in revision.*

---

## Author Response (AR2)

Review "Influence of covariance of aerosol and meteorology on co-located precipitating and non-precipitating clouds over INdo-Gangetic Plains" Gulistan et al.

Thank you for your valuable feedback and continued engagement with our manuscript. We appreciate the time and effort you have invested in reviewing our work. We have carefully considered your comments and would like to address your concerns which led to the improvement of the manuscript. Below are the replies to the reviewer's comments, and indications of additions, modifications, or subtractions to the text under discussion. We report the reviewer's comments in italic black, our responses in italic green, and the text added to the manuscript in roman red.

**Public justification (visible to the public if the article is accepted and published):** *I have read the responses to my review of the article "Influence of covariance of aerosol and meteorology on co-located precipitating and non-precipitating clouds over the Indo-Gangetic Plains". I thank the authors for responding to the various comments, the article is much improved in my opinion.*

- *Thank you for the encouraging remarks and appreciation.*

*Nevertheless, some comments have not been properly answered in my opinion and I would like the authors to consider some changes in the article, which I detail below:*

1. *Regarding the uncertainties, the authors now refer to the various uncertainties in the article, but there is no quantification of the aforementioned uncertainties, nor of the propagation to the results. I am particularly concerned that the observed differences are within the uncertainty range and therefore not significant. Can the authors quantify the uncertainties in their data set? For example, the authors refer to "influence the findings in the current investigation", can they quantify this influence?*

*Reply: Thank you for your valuable suggestion and concern for the quantification of uncertainties. We comply with the expert's valuable suggestion. In this regard the two-sample t test is adopted to characterize the probability 'p' value of datasets at 95% confidence level for MODIS-derived AOD and cloud parameters. Further, figures 8 and 9 are revised as follows for indication of 'p' value in each plot along with the relevant discussion in section 3.4.1 as follows:*

The impact of aerosols on CDNC and CER of PCs and NPCs is illustrated as scatter plots in Fig. 8-9. The quantification of the AOD-CER and AOD-CDNC relationships is demonstrated through detailed linear regressed slopes, regression coefficients ($R^2$) and Pearson's correlation coefficient (R). The color bar represents the variations in LTS. The results show that the two-sample student's t test is carried out to analyze the AOD-CER and AOD-CDNC relationship in view of statistics. The results illustrate that the relationships are statistically significant at 95% ($p < 0.05$) significance level for all study areas.

[Figure]

Fig. 8. AOD-CER and AOD-CDNC regression and correlation coefficient considered at 95% significance level for PCs and NPCs over all study areas in winter season.

[Figure]

Fig. 9. Same as Fig. 8 but in summer season.

2. The table with the number of data points is useful, but does not appear in the article. The number of data is key to the statistical analysis, so I think the reader should have this information. Also regarding the number of data points, the different regimes are associated with relatively low numbers of data. I would like to let the editor judge this and decide whether there are enough data points to infer statistics (a threshold of 15 is applied but is it enough?). However, I was concerned about using the mean instead of the median, as the mean is affected by potential outliers and can be especially important when there is less than 100 points.

*Reply: Thank you for your concern about the significance of our results, as this really led to the improvement of the manuscript. Following is the clarification of your concerns about the data points.*

- *The table with the number of data points is inserted in the supplementary material for the readers.*
- *The Student's t-test is widely used when the sample size is reasonably small (less than approximately 30) (King & Eckersley, 2019). Therefore, two sample student's t test is adopted to compute the value of 'p' at 95% confidence level. In each case it is found to be $p < 0.05$. In this regard Fig. 8-9 are revised (refer to first comment) for indication of 'p' value.*
- *Median is computed instead of mean. In this regard, table 2 in the main manuscript and table 2s in the supplementary material is revised (refer to the following comments).*

3. The dataset colocates cloud and aerosol information based on daily information and does a temporal interpolation using a "statistical function" to fill in data where there is none (aerosol on cloudy day). Can the authors say what the statistical function is? Is it linear? I would also like to see this information in the paper as it is key if someone wants to reproduce the analysis. Also, is there a threshold for the number of days for which AOD is not retrieved? For example, extreme cases: If there is no aerosol information for 30 days, will the authors still collocate an inferred AOD value to colocate with cloud information?

*Reply: We are grateful to reviewer 2 for the careful reading and continued engagement with our manuscript. To accommodate the comment the following text is added in the revised manuscript in the methodology section as follows:*

Subsequently, in this analysis, the VLOOKUP function is utilized for linear interpolation/alignment of the data. This function is available as a built-in feature in Microsoft Excel.

- *Further, if there is no aerosol information then an inferred AOD value is not colocated with cloud information.*

4. Response to the "liquid cloud comment". "Low level clouds can refer to mixed phase or ice clouds, for example in the Arctic and Southern Ocean. I would mention in the article that the study is only about Arctic low-level clouds for readers.

*Reply: Thank you for your insightful comment. Following is the explanation/clarification regarding concerns of data retrieval for liquid clouds.*

- *Since NASA Giovanni is a Web-based application and flexible platform developed by the GES DISC that provides a simple and intuitive way for users to visualize, analyze, and access vast amounts of Earth science remote sensing data over time from the website(https://giovanni.gsfc.nasa.gov ). From this website, the data of liquid and ice cloud parameters can be retrieved separately. Thus we select variables for liquid clouds only e.g., 'liquid water cloud effective particle radius' CER, 'liquid water cloud optical thickness' COT, 'liquid water cloud water path' CLWP etc.*

- *To further confirm/validate that the retrieved data over IGP are for liquid clouds only, we applied the Integrated Multi-satellite Retrievals (IMERG) for Global Precipitation Measurement (GPM) algorithm and retrieved time average maps of precipitation and its probability of liquid phase as follows:*

[Figure]

*Fig. 1. plot (a) shows the precipitation rate in summer from Junly 1-August 31, 2010 and plot (b) shows probability of liquid phase of precipitation.*

[Figure]

*Fig. 2. Probability of liquid phase of precipitation occurred from January 1-31, 2017.*

5. The authors added 3 stations in the eastern part of the region. But they did not respond to my comments as to why they focused on ground-based stations if they did not use measurements from the ground-based instruments. I would have

expected some comparisons with the ground-based instruments, but this is not shown, so I do not understand why they only used satellite information on single pixels. Their number of data points could be greatly improved by using extended regions. I also wonder why the authors did not include information from the southern part? Finally, the 3 stations added in the analysis (on the eastern part) do not appear in the final manuscript (but only in the supporting information), I recommend to include the additional stations on the plots in the main article and maybe moving other stations in the supporting information if the graph is too busy.

*Reply: Thank you for your valuable feedback and precious time. Following is the clarification for this comment.*

- *The present study is not focused on ground-based stations, however, due to the lack of in-situ measuring facilities, and as a result utilization of remote-sensing datasets similar to previous studies (Amin et al., 2009; Nirala et al., 2002), the present study is only satellite-based analysis focused to capture localized variations.*
- *The location of Gandhi College (25.87ºN, 84.13ºE) can be inferred as the southern part of IGP. Further, Delhi is located on the northern edge of IGP extended to the south.*
- *In response to your previous detailed and valuable feedback, we have already tried to include our expanded analysis on the eastern part in the main manuscript. However, it was making the main text too busy. For this reason, it is documented as supplementary material instead.*

6. I thank the authors for responding to my comment about the dependence of the dataset. However, I am not convinced by the answer. My point is that the authors use parameters that are directly derived from the satellite retrievals. Therefore, the dependence between the parameters and the retrievals is subject to the equation and hypothesis that links the parameters on the first basis and not solely from the analysis of the dataset.

*Reply: We are thankful to the reviewer for the comments and suggestions. Per kind suggestions of the reviewer the statistical significance of relationship/dependence between different parameters*

*from different datasets/retrievals is checked through t test at 95% confidence interval (Please refer to comment 1).*

**7.** I agree that MOD06 only refers to TERRA measurements, but I recommend to make it explicit in the text for the reader who is not familiar with the NASA nomenclature.

*Reply: Thanks, and we revised the sentence in the 'Methodology' section as follows.*

This study uses the daily mean of combined dark target and deep blue AOD at 0.55 µm, cloud top pressure (CTP), cloud top temperature (CTT), CF, CER, and COT for liquid clouds from level 3 aerosol-cloud data product MOD08-TERRA.

**8.** From my comment on Table 2, I do not understand why the authors use the mean instead of the median (as I said in a previous comment, the mean is affected by potential outliers, which can have a significant impact with ~70 data points).

*Reply: Thank you for correction. The median values of the respective meteorological parameters are computed. The table 2 and 2S along with relevant discussion are revised as follows:*

The median values computed for the remaining meteorological parameters considered in this study are listed in Table 2. The high values in each case are indicated in bold and low values are italicized. The results show that in winter season the temperature at 850 hPa ($T_{850}$) is relatively high for NPCs ranging from 281 K to 285.6 K. The increase in RH% for PCs during winter ranged from (59.5)% to (71.5)%. Also, the $\Omega > 0$ for NPCs and $< 0$ for PCs in winter season.

In summer season, it is observed that $T_{850}$ is comparatively higher than that for the winter clouds and ranged from 298.3 to 300.2 K and 296.5 to 298.3 K for NPCs and PCs respectively. The high values of $T_{850}$ are due to intense solar fluxes in summer season that keep the temperature of the earth's surface and adjacent atmospheric layer higher. Also, the increase in RH% during summer ranged between 33.5-51.7 % for NPCs. The reason for the high values of wv and RH% is mainly the suitable thermodynamical conditions such as evaporation and convection due to the high temperature of earth surface and air (Sherwood, Roca, Weckwerth, & Andronova, 2010). The results show high values of RH% 70.1% (85%) in winter (summer) season for PCs over Gandhi College. Conversely, notable fluctuations in RH% are observed over the coastal city, Karachi, with values

of 71.5% (65.9%) in winter (summer). Similarly, Fig. 2S and Table 2S show the LTS conditions for PCs and NPCs. The high LTS values indicate more stable condition over Dhaka. Similarly, Table 2S shows the seasonal average values for other meteorological parameters. The results indicate high values of $T_{850}$, RH% and $\Omega$ 295.5 (297.5) K, 88.8 (83.5)% and -0.19 (-0.17) m/s respectively for PCs (NPCs) for over Patna in summer.

Table 12. Meteorological parameters for PCs(NPCs) in summer and winter seasons. Maximum values are for both types of clouds shown in bold and minimum values are indicated as italic.

| | Winter Season | | | Summer Season | | |
|---|---|---|---|---|---|---|
| | $T_{850}$ (K) | RH% | $\Omega$ (m/s) | $T_{850}$ (K) | RH% | $\Omega$ (m/s) |
| Karachi | **284.6 (285.8)** | **71.5** (38) | -0.038 *(0.030)* | *295.9* (298.8) | 65.9 (45.9) | *0.005* (-0.003) |
| Lahore | *280.5 (281.2)* | *59.5* (35.5) | *-0.02* **(0.065)** | **298.3 (300.2)** | 65 *(33.5)* | -0.028 (0.025) |
| Delhi | 284.2 (283.1) | 60.2 *(33.8)* | **-0.1** (0.04) | 296.5 (299.4) | 64.2 (42) | -0.05 *(-0.001)* |
| Kanpur | 283.8 (284.1) | 65.7 (36) | **-0.1** (0.048) | 296.5 (298.4) | 73.7 (43.6) | -0.13 (-0.08) |
| Jaipur | 283.9 (284.1) | 66 (40.5) | -0.065 (0.049) | 296.8 (298.7) | *64* **(51.7)** | -0.04 (-0.029) |
| Gandhi College | 283.2 (284.1) | 70.1 **(45.7)** | **-0.1** (0.05) | 296.9 *(298.3)* | **85** (42.5) | **-0.16 (-0.11)** |

Table 2S. Meteorological parameters for PCs(NPCs) in summer and winter seasons. Maximum values are for both types of clouds shown in bold and minimum values are indicated as italic.

| | Winter Season | | | Summer Season | | |
|---|---|---|---|---|---|---|
| | $T_{850}$ (K) | RH% | $\Omega$ (m/s) | $T_{850}$ (K) | RH% | $\Omega$ (m/s) |
| Kolkata | **286.7 (286)** | 47.4 (39.9) | -0.002 **(0.08)** | *295.5 (295)* | 74.8 (72.8) | -0.15 (-0.14) |
| Dhaka | 285.8 (285.3) | *48.5 (49.2)* | *0.04* **(0.08)** | 294.5 (294.4) | 76.5 (74.6) | -0.13 (-0.10) |
| Patna | 284.7 (284.3) | **64.6 (55.8)** | **-0.06** (0.05) | 295.3 **(297.3)** | **88.8 (83.5)** | **-0.19 (-0.17)** |

9. The authors refer to the "quantification of PDF", what does this mean? Distributions do not tell you whether data sets are similar or not as it depends on the number of data considered, I think only a statistical test can answer that (Student's t-test, Kolmogorov-Smirnov test...).

*Reply: Thank you for the valuable comment and suggestion. Following is the clarification for this comment.*

- *Probability distribution function (PDF) is a normal or Gaussian distribution. It is a type of continuous distribution. Here, the quantification of PDF means to specify the probability of random variable within a particular range of values.*

- *In the current study the PDF is not computed to check the similarity of datasets, but to help visualize the dispersion of a parameter.*

- *Two-sample student's t test is alraedy computed in response to one of your suggestions/comment No.1.*

**Commented [LY1]:** I do not follow this sentence but it seems unnecessary to address the comment and can be deleted. For example,

**Commented [NG2R1]:** This is in response to the question asked in the first sentence of the comment.

**Commented [LY3]:** Better state which suggestion

**Commented [NG4R3]:** I mentioned the suggestion/comment number.

10. l. 282 in the main article with highlighted changes should have units

*Reply: Thank you for the correction. The line 282 is revised as follows.*

The results indicate high values of $T_{850}$, RH% and $\Omega$ are 295.5 (297.5) K, 88.8 (83.5)% and -0.19 (-0.17) m/s, respectively for PCs (NPCs) for over Patna in summer.

References

- King, A. P., & Eckersley, R. (2019). *Statistics for biomedical engineers and scientists: How to visualize and analyze data*: Academic Press.
- Sherwood, S. C., Roca, R., Weckwerth, T. M., & Andronova, N. G. (2010). Tropospheric water vapor, convection, and climate. *48*(2). doi:https://doi.org/10.1029/2009RG000301

- Amin, S. H. A. B. A. N., Crodula, R. O. B. I. N. S. O. N., & Farouk, E. B. (2009). Using MODIS images and TRMM data to correlate rainfall peaks and water discharges from the Lebanese Coastal Rivers. *Journal of Water Resource and Protection*, *2009*.

- Nirala, M. L., & Cracknell, A. P. (2002). The determination of the three-dimensional distribution of rain from the Tropical Rainfall Measuring Mission (TRMM) Precipitation Radar. *International Journal of Remote Sensing*, *23*(20), 4263-4304.

---

## Author Response (AR4)

Review "Influence of covariance of aerosol and meteorology on co-located precipitating and non-precipitating clouds over INdo-Gangetic Plains" Gulistan et al.

We greatly appreciate the time and effort you have invested in reviewing our work. We have carefully considered your comments and addressed your concerns, leading to improvements in the manuscript. Below are our responses to the reviewer's comments, along with indications of additions, modifications, or deletions to the text under discussion. The reviewer's comments are presented in italic black, our responses in italic green, and the text added to the manuscript in roman blue.

**Referee#2 Comments**

- I thank the authors for considering my comments and remarks. I believe the article is improved and ready for publication after minor revisions.
    - *Thank you so much. Your appreciation is always an honor.*

1. Table 2 caption: Can the authors state in the caption that it is the median?

*Reply: Thanks for the kind suggestion. The caption is revised as follows in the revised manuscript:*

**Table 1.** Median values of meteorological parameters for PCs(NPCs) in summer and winter seasons. Maximum and minimum values for both types of clouds are shown in bold and *italic*, respectively.

2. VLOOKUP: The authors mentioned that they used the VLOOKUP function to interpolate the AOD. I am not familiar with this function, and after a search on the internet, VLOOKUP does not seem to interpolate values, it can (at best) fill in missing values.

*Reply: Thanks for correction. The text is revised as follows:*

The VLOOKUP function in Microsoft Excel is applied to filtering out counts where data is not available, searching for values of a parameter in the first column, and retrieving the values of other parameters in the same rows on the corresponding dates in a large dataset.

3. Can the authors explain how interpolation works?

*Reply: Interpolation of satellite data is the process of estimating values between measured data points captured by satellites. Since satellite data is often collected at discrete points or intervals,*

*interpolation helps to create a continuous dataset by predicting values at locations where direct measurements are not available. This is commonly used to improve the resolution of the data and to fill gaps, making the data more useful for various analyses and applications such as weather forecasting, climate monitoring, and environmental studies.*

*However, in this study, interpolation techniques were not applied due to limitations of data and was mentioned in the manuscript mistakenly.*

4. The low cloud is only liquid. I completely agree with the authors. However, I think it would help the reader if it was explicitly mentioned in the text that low-level clouds are liquid.

*Reply: Thank you so much. The valuable suggestion is incorporated in the text in line no. 92-94 of the introduction section of the revised manuscript as follows:*

Therefore, the present study aims to deepen the previous study (Anwar et al., 2022), by focusing on the ACPI for low-level liquid clouds.

*The authors sincerely appreciate the valuable time, insightful feedback, and continued engagement of the esteemed reviewer with our manuscript, based on your input and useful comments, the quality of the manuscript is improved significantly.*
* * *
**Referee #3 Comments**

The present study analyzed the response of liquid clouds, including PCs and NPCs in winter and summer to aerosol loadings in six various cities over the Indo-Gangetic Plains (IGP) under a variety of meteorological conditions. The dataset used include the long-term 21 (2001-2021) retrievals from Moderate Resolution Imaging Spectroradiometer (MODIS), Tropical Rainfall Measuring Mission (TRMM), and National Center for Environmental Prediction/National Center for Atmospheric Research (NCEP/NCAR) reanalysis-II datasets. The authors attempted to

quantify the ACPI by investigating the covariance analysis methods between meteorology, AOD and cloud properties.

The methods are not novel, but the findings are not convincing to me, and further clarification and discussion are needed before recommending its acceptation.

*Reply: We appreciate the insightful review of the manuscript and the concerns regarding the novelty of our methodology and findings. While we acknowledge that the methodology is not completely novel, however, this study presents several unique and convincing findings that have not been addressed in previous research. A few of these findings, along with the innovative aspects of our methodology, are outlined below:*

*IGP is uniquely significant for aerosol and cloud studies due to its high population density, extensive agricultural activities, and rapid industrialization. The unique meteorological conditions, including the monsoon system and seasonal variations, provide a natural laboratory for observing how aerosols influence cloud formation, precipitation patterns, and atmospheric dynamics. However, to date, none of the studies reported such a long-term and comprehensive analysis using both satellite observations and reanalysis data distinguishing between precipitating and non-precipitating clouds across this region which is heavily impacted by human activities. Additionally, this study presents several unique and convincing findings that have not been addressed in previous research. A few of these are outlined below:*

- *A decadal decrease in aerosols is observed over the coastal city, Karachi, and the less industrialized city, Jaipur. Whilst the increase in high percentage is observed over Ghandi College.*
- *The high frequency of cloudy days and precipitating clouds observed over Ghandi College in the summer season indicates a higher moisture content in the atmosphere during the summer at this location, leading to more cloud formation and precipitation.*
- *Values of CF for low-level clouds (both PCs and NPCs) were found to be higher during the winter season. This can be due to lower temperatures and increased atmospheric stability*

- *Whilst the favorable meteorological conditions such as higher temperatures and increased humidity, lead to more convection and cloud formation, resulting in higher precipitation rates in the summer season.*
- *During summer, Karachi experienced the most stable meteorological conditions (LTS=15.9±0.87 K), resulting in less precipitation.*
- *While Ghandi College exhibited the most unstable conditions (LTS=9.6 ±0.4 K). Ghandi College's low LTS values suggest it experiences more turbulent and convective weather, leading to more cloud formation and precipitation.*
- *AOD exhibited a strong correlation with CER for PCs. A strong correlation between AOD and CER in PCs suggests that aerosols influence the microphysical properties of clouds, affecting droplet size and precipitation processes.*
- *Similarly, the good correlation of AOD with CDNC in NPCs indicates that aerosols play a significant role in cloud formation and characteristics, affecting cloud reflectivity and lifetime.*

The major concern and comments are listed as below:

**Major comments:**

1. It occurs to me that there are not definitions for the PCs and NPCs in this manuscript, which are of greater importance to the results interpretation. Also, the motivation for this comparison analysis is suggested to be clarified. For instance, Why not focus on the warm clouds in terms of initial stage (cumulus) versus mature stage?

*Reply: Thank you for your valuable feedback, insightful comments, and precious time as this led to the improvement of the manuscript. Following is the clarification/explanation. The following text is inserted in line 51 in the revised manuscript:*

Precipitating clouds are thick clouds with significant vertical development and high moisture content, forming under unstable atmospheric conditions, such as cumulonimbus and nimbostratus, that produce precipitation reaching the ground. In contrast,  non-precipitating clouds are typically

thin, have low moisture content, and form under stable atmospheric conditions, including cloud types like cirrus, cirrostratus, altostratus, and stratus, which generally do not produce significant precipitation (Houze Jr, 2014).

Houze Jr, R. A. Nimbostratus and the separation of convective and stratiform precipitation. In International geophysics. Elsevier. 104, 141-163, 2014.

*We sincerely appreciate and acknowledge the kind suggestion of the respected reviewer to focus on the warm clouds in terms of the initial stage (cumulus) versus the mature stage. That involves the analysis of the temporal and spatial evolution of cumulus clouds from the initial into the mature stage and the investigation of associated variations in cloud properties and precipitation patterns. However, we cannot carry out this study because of data limitations. Further, as mentioned in the second last paragraph of section 1 in the manuscript, the motivation for this work is to deepen one of our recent studies (Anwar et al., 2022) through a long-term and detailed analysis including other significant meteorological parameters such as LTS, PR, and $T_{850}$, cloud microphysical parameter CDNC, categorization of clouds into PCs and NPCs and extension from monsoon regions of Pakistan to other locations of Indo-Gangetic Plains.*

2. The caveat for the unclear definition of CP and NCP contains the seemly contradictory results and findings: e.g., L32-35: I can not understand the findings such as: (2) The AOD-CER correlation is good (weak) for precipitating clouds (PCs) but weak for non-precipitating clouds (NPCs); (2) The sensitivity value of the first indirect effect is high for for PCs and low for NPCs. As a fact of matter, the precipitation stage is less susceptible to aerosol effect compared with cloud stage. Besides, the logic behind the aerosol effect on precipitation, if any, goes like: "By radiative or microphysical effect, aerosol firstly affects the properties of clouds, thereby influencing the formation and evolution of precipitation". I am surprised to see the above-mentioned two main findings that are out of the scope of my physical understanding for this issue. It should be clarified from the perspective of physical process.

*Reply: Thank you for your concern about our results. Following is the clarification of your concerns about our results and findings:*

- *The two results are interlinked as follows:*

*when we say, "The sensitivity value of the first indirect effect is high for PCs," it signifies that in favorable atmospheric conditions, the moisture readily condenses on existing aerosol particles in PCs. This leads to the formation of numerous smaller cloud droplets with high concentrations. As these smaller droplets collide and coalesce, they grow to larger size. Consequently, this phenomenon results in a strong AOD-CER correlation for PCs, and a weak for NPCs.*

3. Why the authors only show the results for 6 different cities over the IGP but not the overall region of IGP. As the dataset used basically are acquired from satellite, it will not be difficulty to accomplish this task. Therefore, for the benefit of readers to gain a full understanding of ACPI in this region, I strongly suggest the authors add the overall analysis over the whole region of IGP.

*Reply: We appreciate the esteemed reviewer's valuable suggestion. The Indo-Gangetic Plain (IGP) is characterized by a diverse and extensive pool of aerosols due to its unique topography, making the entire region crucial for analyzing ACPI. The Himalayas to the north act as barriers to the winds, leading to the trapping of aerosols over the central part of the IGP, resulting in a high concentration of anthropogenic aerosols and creating a dense aerosol belt at the base of the mountains.*

*In the earlier version of the manuscript, we focused on six key locations. Following the honorable reviewer's recommendation in the first revision, we extended our study area to the eastern part of the IGP by including three additional sites. To avoid overcrowding the manuscript and due to the similarity of the results, we have documented these findings in the supplementary material, as noted in the last line of section 2.1, "Study Area."*

*We deeply respect the valuable feedback and kind suggestions of our esteemed reviewer to expand it further. In light of this, we plan to expand our study to encompass the southern part of the IGP in future research. We believe this will provide a more comprehensive analysis and further enrich our understanding of the region*

4. Section 2.1 & Figure 1: The captioon of Figure should be expanded to include more information, such as the city marked by blue dots. Besides, the legend for these blue dots

are missing. Lastly, the upper, middle, and eastern portions of the IGP are suggested to be marked on this figure.

*Reply: Thanks for the comments. Per your valuable suggestion, more information is added and is inserted in the revised manuscript as follows:*

[Figure]

5. It occurs to me that the title emphasize the contrast of aerosol impact on PC versus NPC, but there are not any key points in Highlights being specific for the aerosol impact on either PC or NPC. Why?

*Reply: Thanks for pointing this out. The 'highlights" are accordingly revised as follows:*

Strong aerosol-cloud interactions under unstable meteorological conditions led to the formation of thick precipitating clouds.

6. The English writing should be improved.

*Reply: Thanks; We have tried our best.*

**Minor comments:**

1. L19-21: RETRIEVALS can be used to describe satellite product, but not for reanalysis, and thus the authors are suggested to rephrase it.

*Reply: Thanks, the valuable suggestion is implemented, and the text is revised as follows in the revised manuscript.*

Therefore, this study aims to systematically investigate the effects of aerosols and meteorological factors on ACPI in the co-located precipitating (PCs) and non-precipitating clouds (NPCs) clouds in winter and summer seasons by analyzing the long-term (2001-2021) retrievals from Moderate Resolution Imaging Spectroradiometer (MODIS), and Tropical Rainfall Measuring Mission (TRMM) products coupled with the National Center for Environmental Prediction/National Center for Atmospheric Research (NCEP/NCAR) reanalysis-II datasets over the Indo-Gangetic Plains (IGP).

2. L38-39: Does "the precipitation rate (PR) exhibits high values in summer season" have any logic correlation with "and PR values are found high in comparatively thin clouds with fewer CDNC.."?? why did the authors put these two sentences together into one sentence???

*Reply: Thanks for the correction. To accommodate the comment and further clarify, the text is revised as follows in the revised manuscript:*

Furthermore, the precipitation rate (PR) exhibits high values in the summer season, primarily due to the significant contribution from optically thick clouds with lower CDNC ($< \sim 50$ cm$^{-3}$) and larger CER and intermediate contribution from optically thick clouds with higher CDNC ($> \sim 50$ cm$^{-3}$).

3. L45-47: ACPI is generally referred to aerosol indirect effect, whereas ARI refers to aerosol direct effect. However, the authors use both terms cursively, which will make the readers and me confused. Therefore, the authors can rephrase it to make the expression more consistent throughout the whole manuscript.

*Reply: Thank you for your precious time. The valuable suggestion is implemented.*

4. L68: Grammatic erros in "and makes".

*Reply: Sorry for the typo and we fixed it.*

5. L69-71: is "heat wave" associated with the topic investigated in this manuscript? Also, necessary references are needed to support "frequent variations in cloud fraction (CF), extreme precipitation…"

*Reply: Thank you for the valuable suggestion. The clarification/explanation is as follows.*

- *The interactions between aerosols, clouds, and precipitations are complex and can have varying impacts on heat waves depending on the types of aerosols, cloud properties, and regional climatic conditions.*
- *The following reference is added at the end of the sentence as follows in the revised manuscript.*

Besides, frequent variations in cloud fraction (CF), extreme precipitation, abrupt temperature changes (e.g., heat waves), and irregular unseasonal rains may cause major hazards at local and regional levels in the future (Zhou et al., 2020).

Zhou, S., Yang, J., Wang, W., Zhao, C., Gong, D., Shi, P.: An observational study of the effects of aerosols on diurnal variation of heavy rainfall and associated clouds over Beijing–Tianjin–Hebei, Atmos. Chem. Phy., *20*, 5211-5229,https://doi.org/10.5194/acp-20-5211-2020, 2020.

6. L72-73: Some important work in the literature is missing for the ACI from observations, including but not limited to: https://doi.org/10.1126/science.1089424; https://doi.org/10.1016/j.atmosenv.2015.04.063 ; https://doi.org/10.1029/2019GL085442.

*Reply: Thank you for suggesting very informative and valuable research work. To accommodate the kind suggestion, we added this and a few more references in the revision as follows::*

In the last two decades, the scientific community has focused on the quantification of ACI using both observations (Feingold et al., 2003; Koren et al., 2004; Costantino et al., 2010; Wang et al., 2015; Zhao et al., 2018, Guo et al., 2019; 2020; Anwar et al., 2022)

Guo, J., Su, T., Chen, D., Wang, J., Li, Z., Lv, Y., Zhai, P.: Declining summertime local-scale precipitation frequency over China and the United States, 1981–2012: The disparate roles of aerosols. *Geophy.Res.Letrs.*, *46*(22), 13281-13289, https://doi.org/10.1029/2019GL085442 ,2019.

Koren, I., Kaufman, Y. J., Remer, L. A., & Martins, J. V.: Measurement of the effect of Amazon smoke on inhibition of cloud formation. Sci., 303(5662), 1342-1345, https://doi.org/10.1126/science.1089424, 2004.

Wang, F., Guo, J., Zhang, J., Huang, J., Min, M., Chen, T., Li, X.: Multi-sensor quantification of aerosol-induced variability in warm clouds over eastern China. *Atmos. Envi.*, *113*, 1-9, https://doi.org/10.1016/j.atmosenv.2015.04.063 , 2015.

7. L98: w rather than Ω, is used to represent vertical velocity.

*Reply: Thank you. The good suggestion is implemented.*

8. There are too many grammar errors or inappropriate expressions throughout this manuscript, and I strongly suggest the authors find the help to touch up this work, at least from the perspective of English writing. Sorry that I can not be more positive at the current stage.

*Reply: Thank you. We have tried our best and got professional help with English.*

*We would like to extend our sincere gratitude to the esteemed reviewer for their insightful and constructive feedback. Based on your constructive comments, the quality of the manuscript is significantly improved.*
* * *

---

## Author Response (AR5)

Review "Influence of covariance of aerosol and meteorology on co-located precipitating and non-precipitating clouds over INdo-Gangetic Plains" Gulistan et al.

Thank you for your thoughtful feedback and for taking the time to engage with our manuscript. We greatly appreciate the effort you've put into reviewing our work. We have carefully reviewed your comments and addressed your concerns, resulting in significant improvements to the manuscript. Below are the replies to the reviewer's comments, and indications of additions, modifications, or subtractions to the text under discussion. We report the reviewer's comments in italic black, our responses in italic green, and the text added to the manuscript in Roman blue.

**Referee #1 Comments (Minor Revisions)**

I thank the authors for their efforts to change the manuscript. I do not have more comment that I already made. But I would like to highlight two points:

1.  Reading reviewer 3's comment #3, I noticed that he also wonders why the analysis focuses only on the stations and not on the whole regions, since the datasets used by the authors do not depend on ground-based measurements. I have already made this comment (comment 5 on my review of 23.02.0224), it is still not clear in the article why the authors focus on specific sites rather than the whole region.

*Reply: The authors acknowledge the validity of this concern and are grateful for your persistence in seeking clarity. For clarification in the article the following text is inserted in line no 99 as follows:*

Due to the absence of in-situ measurement facilities and the constraints of limited computational resources, the study concentrated on satellite data for specific locations across the entire IGP. These locations were strategically chosen due to their positioning within significant aerosol belts, where the concentration and behavior of aerosols are of particular interest. Therefore, the satellite-based approach was chosen as it provides detailed insights into aerosol dynamics in these critical regions while also benefiting from the broader spatial coverage of satellite data.

2. I am still concerned about the number of data points and wonder if conclusions can be drawn from statistics with less than 50 datapoints, most of the time (as I mentioned in my comment #6 of 23/02/2024). Even if the statistical test gives a result, it does not mean that it is robust. I will specifically point this out to the editor.

*Reply: Thank you for your continued feedback and for raising your concerns regarding the sample size. We understand the importance of having enough data points for statistical robustness. However, our study aimed at identifying preliminary trends and patterns that could inform future research. We have also acknowledged this limitation of our study in the 'Conclusion' section in the manuscript and have positioned our findings as a foundation for further, more extensive studies by inserting the following text in line 613:*

Although the sample size limits the study, the observed trends offer important insights that provide a foundation for future research. Therefore, further investigations with larger sample sizes are suggested to validate and extend these findings.

*We sincerely appreciate the thoughtful and constructive feedback provided by the esteemed reviewer.*

*\*\*\*\*\*\*\*\*\*\*\*\*\*\*\*\*\*\*\*\*\*\*\*\*\*\*\*\*\*\*\*\*\*\*\*\*\*\*\*\*\*\*\*\*\*\*\*\*\*\*\**